# An adaptable, reusable, and light implant for chronic Neuropixels probes

Célian Bimbard[1]*, Flóra Takács[2], Joana A Catarino[3], Julie MJ Fabre[4], Sukriti Gupta[5], Stephen C Lenzi[2], Maxwell D Melin[6], Nathanael O'Neill[4], Ivana Orsolic[2], Magdalena Robacha[1], James S Street[4], José M Gomes Teixeira[3], Simon Townsend[7], Enny H van Beest[1], Arthur M Zhang[8], Anne K Churchland[6], Chunyu A Duan[2], Kenneth D Harris[4], Dimitri Michael Kullmann[4], Gabriele Lignani[4], Zachary F Mainen[3], Troy W Margrie[2], Nathalie L Rochefort[8,9], Andrew Wikenheiser[5], Matteo Carandini[1], Philip Coen[1,10]*

[1]UCL Institute of Ophthalmology, University College London, London, United Kingdom; [2]Sainsbury Wellcome Centre for Neural Circuits and Behaviour, University College London, London, United Kingdom; [3]Champalimaud Research, Champalimaud Centre for the Unknown, Lisbon, Portugal; [4]UCL Queen Square Institute of Neurology, University College London, London, United Kingdom; [5]Department of Psychology, University of California, Los Angeles, Los Angeles, United States; [6]Department of Neurobiology, University of California, Los Angeles, Los Angeles, United States; [7]The FabLab, Sainsbury Wellcome Centre for Neural Circuits and Behaviour, University College London, London, United Kingdom; [8]Centre for Discovery Brain Sciences, School of Biomedical Sciences, University of Edinburgh, Edinburgh, United Kingdom; [9]Simons Initiative for the Developing Brain, University of Edinburgh, Edinburgh, United Kingdom; [10]Department of Cell and Developmental Biology, University College London, London, United Kingdom

*For correspondence:
c.bimbard@ucl.ac.uk (CB);
p.coen@ucl.ac.uk (PC)

Competing interest: The authors declare that no competing interests exist.

## eLife Assessment

This **valuable** study presents the design of a new device for using high-density electrophysiological probes ('Neuropixels') chronically and in freely moving rodents. The evidence demonstrating the system's versatility and ability to record high-quality extracellular data in both mice and rats is **compelling**. This study will be of significant interest to neuroscientists performing chronic electrophysiological recordings.

**Abstract** Electrophysiology has proven invaluable to record neural activity, and the development of Neuropixels probes dramatically increased the number of recorded neurons. These probes are often implanted acutely, but acute recordings cannot be performed in freely moving animals and the recorded neurons cannot be tracked across days. To study key behaviors such as navigation, learning, and memory formation, the probes must be implanted chronically. An ideal chronic implant should (1) allow stable recordings of neurons for weeks; (2) allow reuse of the probes after explantation; (3) be light enough for use in mice. Here, we present the 'Apollo Implant', an open-source and editable device that meets these criteria and accommodates up to two Neuropixels 1.0 or 2.0 probes. The implant comprises a 'payload' module which is attached to the probe and is recoverable, and a 'docking' module which is cemented to the skull. The design is adjustable, making it easy to change the distance between probes, the angle of insertion, and the depth of insertion. We tested the implant across eight labs in head-fixed mice, freely moving mice, and freely moving rats. The number of neurons recorded across days was stable, even after repeated implantations of the

same probe. The Apollo implant provides an inexpensive, lightweight, and flexible solution for reusable chronic Neuropixels recordings.

## Introduction

Some fundamental cognitive processes develop across days (e.g. learning) and are best studied in naturalistic environments (e.g. navigation). To gain insights into these processes, it is necessary to record brain activity chronically and to be able to do so in freely moving animals. Chronic recordings in freely moving animals are possible with calcium imaging (*Ghosh et al., 2011*; *Zong et al., 2022*). However, accessing deep brain regions can require invasive surgery and fails to capture fast neural dynamics. Electrophysiology overcomes these issues: the temporal resolution is higher, deeper regions are readily accessible, and recordings can be made in freely moving animals. Substantial effort has thus been dedicated to developing devices for chronic electrophysiology recordings (*Berényi et al., 2014*; *Chung et al., 2017*; *Chung et al., 2019*; *Ferguson et al., 2009*; *Ferreira-Fernandes et al., 2023*; *Newman et al., 2023*; *Okun et al., 2016*; *Schoonover et al., 2021*; *Shobe et al., 2015*). But these devices are typically non-recoverable, are too heavy for use in smaller animals like mice, or record relatively few neurons.

With Neuropixels probes, many hundreds of neurons can be recorded in a single insertion (*Jun et al., 2017*; *Steinmetz et al., 2021*). These probes allow experimenters to produce brain-wide maps of neural activity in head-restrained mice using acute recordings (*Allen et al., 2019*; *Benson et al., 2023*; *Steinmetz et al., 2019*; *Stringer et al., 2019*). To track neurons across days, and to use freely moving animals, the probes can be implanted chronically, with procedures that are permanent (*Jun et al., 2017*; *Steinmetz et al., 2021*) or recoverable (*Ghestem et al., 2023*; *Horan et al., 2024*; *Juavinett et al., 2019*; *Luo et al., 2020*; *Song et al., 2024*; *Steinmetz et al., 2021*; *van Daal et al., 2021*; *Vöröslakos et al., 2021*). Permanent implants are lightweight and stable, but their use at scale is not financially feasible. Conversely, recoverable implants can be reused, but solutions need to be cheaper, lighter, more flexible, and easier to implant and explant. In particular, the only published recoverable implants for Neuropixels 2.0 probes may be too heavy for use with typical mice, and cannot be adjusted for different implantation trajectories (*Steinmetz et al., 2021*; *van Daal et al., 2021*).

To address these issues, we developed the 'Apollo implant' for the reversible chronic implantation of Neuropixels probes. The implant is named for its lunar module design: a recoverable payload module accommodates up to two Neuropixels probes and is reused across animals, and a docking module is permanently cemented to the skull during implantation. The design is open source and can be readily adjusted with editable parameters to change distance between probes, implantation depth, or angle of insertion.

Our eight independent laboratories have performed successful recordings with the Apollo implant in mice and rats, supporting the flexibility and simplicity of the design. The same Neuropixels probes have been reimplanted up to six times with no significant change in recording quality. Recordings were stable across weeks and sometimes months. This allows for recordings to cover the entirety of the probes (by recording from different sections across days), while minimizing setup time, and could facilitate the tracking of neurons across days. The design has been independently printed, adjusted, and implanted across labs, and implanted subjects included freely behaving mice and rats and head-fixed mice, with Neuropixels 1.0, 2.0α (a pre-release version), and 2.0 probes.

## Results

### Flexible design

The Apollo implant consists of two parts, the payload and the docking modules, inspired by previous designs (*van Daal et al., 2021*; *Figure 1*). Both parts can be 3D-printed in a variety of materials, although we typically used a combination of Nylon PA12 and Formlabs Rigid Resin. The Neuropixels 1.0, 2.0α, and 2.0 implants weigh 1.7, 1.3, and 0.9 g (*Table 1*). Payload modules can accommodate up to two parallel probes, with the second probe adding a further 0.4, 0.2, and 0.2 g. The Apollo implant is therefore 40% lighter than the only published Neuropixels 2.0α solution, which recommends animals are at least 25 g (*van Daal et al., 2021*; *Steinmetz et al., 2021*). This allows for use in

**eLife digest** Certain cognitive processes, such as learning, develop over relatively long periods (hours or days). Others, including navigation – the ability to move through a space based on our knowledge of it – usually take place when the subject is free to explore its environment. This can make studying these processes challenging, as researchers need to record brain activity for long periods and in freely moving subjects.

Electrophysiology allows researchers to record brain activity at the millisecond timescale, but technical constraints have made it difficult to record more than a few neurons for any length of time. However, a set of electrophysiology probes called Neuropixels have been developed to allow the recording of hundreds of neurons at once. These probes can be permanently implanted in the brain to track neural activity over long periods. Unfortunately, these implants make it impossible to recover the probes, making their use too expensive for most researchers.

To address this issue, Bimbard et al. set out to develop an implant that would allow the reversible implantation of Neuropixel probes, allowing researchers to track hundreds of cells at fast timescales and over long periods. The device they developed, called the Apollo implant, is a lightweight, reusable device with an open-source design that can be adjusted to suit experimental needs.

Bimbard et al. combined data from eight independent laboratories using the Apollo implant to demonstrate that it can be easily reproduced and modified. These data show that the implant can measure neural data stably for over 100 days after the initial implantation. Additionally, Bimbard et al. show that it is possible to reimplant the same probes many times without losing recording quality.

The Apollo implant makes long-term tracking of groups of neurons reliable and affordable, which will facilitate cognition studies across different model systems.

smaller animals (e.g. female and water-restricted mice). The electronics are protected by lids with slots to accommodate the flex cables when not connected to the headstage (*Figure 1A*). To ensure the implant is maximally compact, flex cables can be folded into the cavity beneath the lids (*Figure 1B*). This minimizes implant height (29, 21, and 17 mm for a Neuropixels 1.0, 2.0α, and 2.0), reducing the moment of inertia above the head. The implant can be 3D-printed for $10 ($3 for each disposable module).

The implant is flexible and recoverable, allowing for different configurations, and the same Neuropixels probe(s) to be used multiple times (*Figure 1C*). Once the payload module is constructed, the distance between the two probes remains fixed. The docking module is connected to the payload module via small screws, which makes it easy to assemble, and disassemble upon explantation. Only the docking module is cemented to the animal's skull, and it is covered with merlons to increase contact with the cement and therefore the stability of the implant. To facilitate different implantation depths and angles with the same payload module, both the length and base-angle of the docking module can be adjusted. The base of the implant thus remains parallel to the skull (*Figure 1C*) which improves stability and reduces weight by minimizing implant height and the quantity of cement required. All adjustments can be achieved by inexpert CAD users with preset parameters supplied in the editable files to change distance between probes (1.8–6.5 mm—beyond 6.5 mm, two implants can be used), implantation depth (2–6.5 mm), or angle of insertion (up to 20 degrees) (*Video 1*). As the fully editable files are provided, users can (and have) adjusted the implants to exceed these default boundaries, or create their own custom modifications which are also available online (see Methods).

To help combine the payload and docking modules, we designed a dedicated constructor (*Figure 1D*). The docking holder, containing a new docking module, slides onto the constructor posts, and the payload holder, containing the payload module, is fixed to the end. The two modules are thus coaxial, and the docking module can slide into position and be secured to the payload module without risk of damaging the probes. The constructor comprises 3D-printed parts and Thorlabs 6 mm poles for a one-time cost of $25.

## Assembly and implantation

A comprehensive protocol for assembly and implantation, including variations employed across labs, is provided in Methods. Payload modules are assembled with one or two Neuropixels probes. After

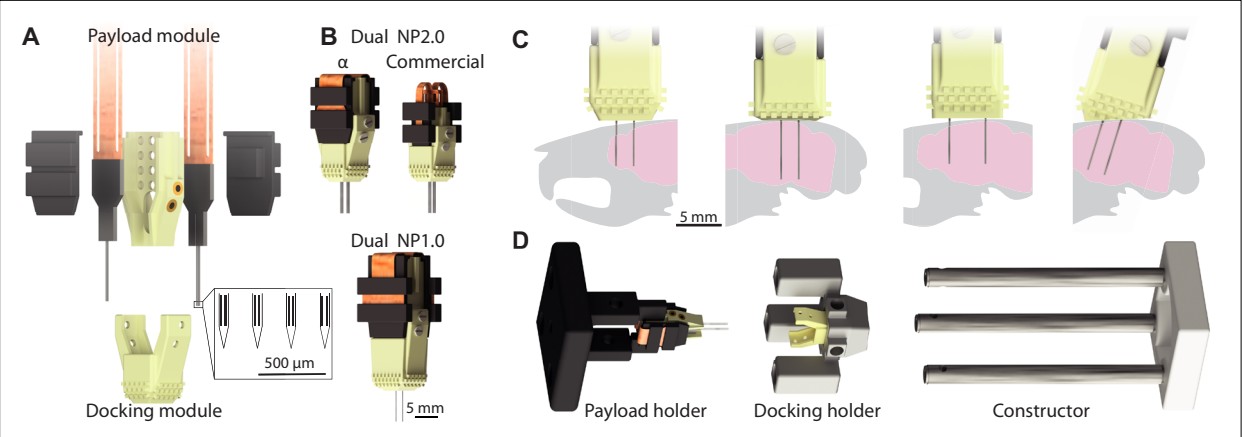

**Figure 1.** The Apollo implant and its flexible design. (**A**) Exploded view of the implant showing the two modules: the payload module, which accommodates up to two Neuropixels probes (protected by two lids), and the docking module. Zoom-in: scaled illustration of the tip of a 4-shank Neuropixels 2.0α probe. Each shank is 75 µm wide, with 250 µm center–center distance between shanks and 15/32 µm vertical/horizonal distance between electrode sites on each shank. (**B**) Assembled view of the implant, for 2.0α and 2.0 (top) and 1.0 (bottom) probes. (**C**) Illustration of implant flexibility. Compared with the standard model (left), the length of exposed probes (middle-left), spacing between probes (middle-right), and implantation angle (right) can all be adjusted with preset parameter changes in the software files (**Video 1**). (**D**) Constructor for the assembly of the payload and docking modules. The docking holder slides along the posts of the constructor, and optimally aligns with the payload module being held by the payload holder. This effectively eliminates the risk of breaking the shanks when combining modules.

probe-sharpening (see Methods, *Figure 2—figure supplement 1A*), an empty payload module was positioned on adhesive putty and coated with a thin layer of epoxy. The probe(s) can then be affixed and aligned to the payload module, either by eye or using graph paper, before covering the base and electronics with epoxy or dental cement (*Figure 2A*). The flex cable was folded and inserted into the lid and lids were then glued to the payload module (*Figure 2B*).

The payload module (new or previously used) was combined with a new docking module for each experiment. Docking modules were adjusted to match experimental requirements (e.g. insertion depth, angle, etc.). The docking module was secured in its holder and slid onto the arms of the constructor. The payload module was secured in its holder and attached to the end of the constructor (*Figure 2C, D*). The docking module holder was then slid along the constructor arms, and the two modules were secured with screws (*Figure 2D*). Before each experiment, any gaps in the assembled

**Table 1.** Implant weight depends on probe version and material.

The weight for each implant version. We find these to vary with each print (5%), and with different services (10–15%). For consistency, these weights are calculated from part volume and the material density. The 'Standard' implant comprises PA12-Lids and Rigid4000-Payload/Docking modules. This is the most common implant used by experimenters, but the PA12-only implant has been used to reduce weight further. The total weights do not include the cement (0.2 g) to fix the probes to the payload module.

| | Weights of implants (g) | | | | | |
|---|---|---|---|---|---|---|
| | NP 1.0 | | NP 2.0-Alpha | | NP 2.0-Commercial | |
| | Nylon PA12 | Rigid4000 Resin | Nylon PA12 | Rigid4000 Resin | Nylon PA12 | Rigid4000 Resin |
| Payload | 0.36 | 0.46 | 0.29 | 0.38 | 0.18 | 0.23 |
| Docking | 0.35 | 0.45 | 0.26 | 0.34 | 0.22 | 0.29 |
| Lid (x2) | 0.61 | 0.79 | 0.41 | 0.52 | 0.25 | 0.32 |
| Probe (x2) | 0.80 | | 0.38 | | 0.33 | |
| Screws (x4) | 0.09 | | 0.09 | | 0.09 | |
| Threads (x4) | 0.08 | | 0.08 | | 0.08 | |
| PA12 total | 2.29 | | 1.51 | | 1.15 | |
| Standard total | 2.50 | | 1.67 | | 1.26 | |

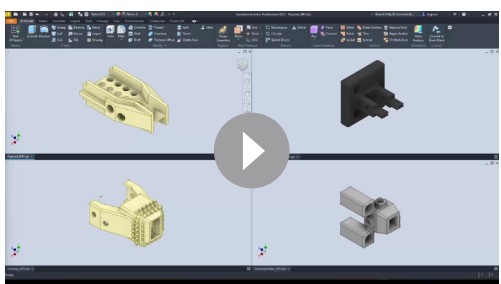

**Video 1.** Guide to altering the Apollo parts to a particular implantation. This video guide demonstrates how to change the shape of the implant using parameters in Autodesk Inventor software—facilitating changes in inter-probe distance, penetration depth, and angle of implantation.

https://elifesciences.org/articles/98522/figures#video1

implant were filled (*Figure 2E*). Prior to each implantation, probes were typically coated with fluorescent dye for post-experiment trajectory tracking (*Figure 2—figure supplement 1B*).

Craniotomies were performed on the same day as the implantation, but this could be any time after assembly (*Figure 2F*). The implant was held using the 3D-printed payload holder and positioned using a micromanipulator. The eight shanks (in the case of a dual 4-shank implant) are positioned at the surface of the brain (*Figure 2G*). Care is taken to avoid large blood vessels, and the implant can be rotated and repositioned. If the vessel is not completely avoidable, the shanks can be positioned on each side of the blood vessel. Probes were inserted to the desired depth at a slow speed (3–5 μm/s). Finally, to complete the implantation, the docking module was cemented to the skull (*Figure 2H*).

## Explantation

Explantations were performed with a payload holder attached to a micromanipulator. The holder was aligned to the payload module, slid into place, and secured with a screw. The screws between the payload and the docking modules were then removed, and the payload module extracted (*Figure 2—figure supplement 1B*). Probes were cleaned with a Tergazyme solution, occasionally followed by a silicone cleaning solvent if Dural-Gel stuck to the probe. The payload module was combined with a new docking module for subsequent experiments.

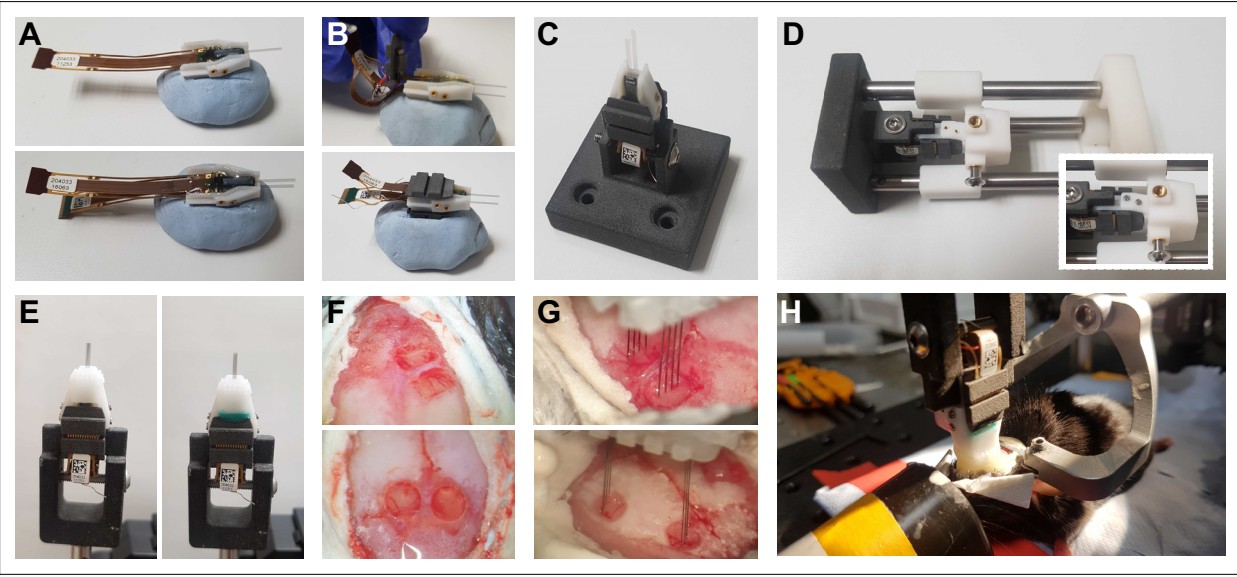

**Figure 2.** Assembly and implantation. (**A**) Initial stage of payload assembly. The payload module is stabilized on Blue Tack while the first (top) and second (bottom, optional) probes are secured with epoxy. (**B**) Each flex cable is first folded into a cavity in the lid (top) before the lid is glued in place (bottom). (**C**) The completed payload module fixed in its holder before being attached to the constructor. (**D**) The combination of payload and docking modules in the constructor. Inset: after the screws have been added to combine the modules. (**E**) Before (left) and after (right) residual gaps were filled with Kwik-Cast. (**F**) Example of dual craniotomies performed with a drill (top – premotor cortex and striatum) or biopsy punch (bottom – bilateral superior colliculus). (**G**) Dual 4-shank probes at the initial stage of insertion into craniotomies performed with drill (top) or biopsy punch (bottom). (**H**) Finalized implant in anesthetized animal, after the docking module has been cemented to the skull.

The online version of this article includes the following figure supplement(s) for figure 2:

**Figure supplement 1.** Additional protocol steps.

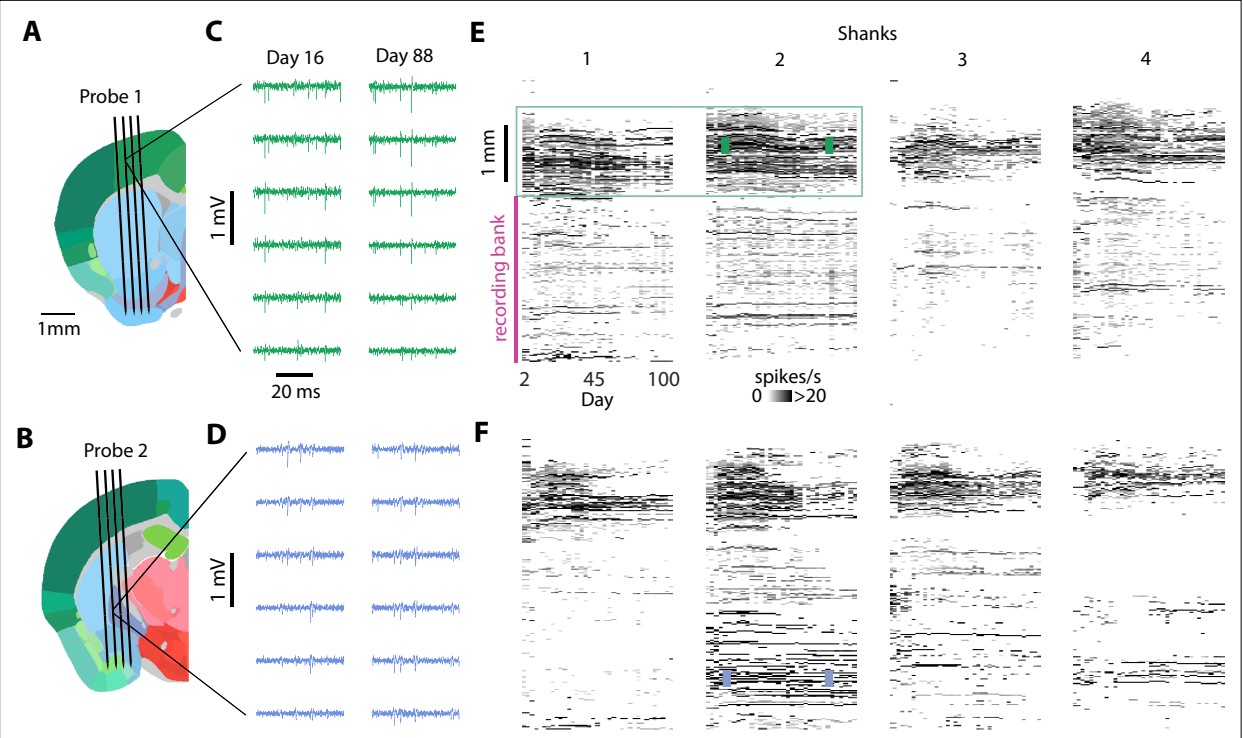

**Figure 3.** Dual implant providing months of recordings. (**A, B**) Insertion trajectories of two simultaneously implanted 4-shank Neuropixels 2.0α probes, with respect to brain anatomy (Allen Mouse Brain Atlas, *Wang et al., 2020*). (**C, D**) Raw signal (bandpass filtered between 400 Hz and 9 kHz) across six channels, on day 16 and 88 post-implantation. (**E, F**) Number of spikes per second versus depth along the probe (*y*-axis) and days from implantation (*x*-axis) for the same implantation shown in A–D. The total number of spikes per second (across all detected units) is binned across depths for each day (20 μm bins). This mouse was recorded while head-fixed.

Across laboratories, 97% of probes were recovered without any broken shanks (61/63 explanted probes, *Supplementary file 1*). In only two cases were probes damaged, and in one of those cases the skull integrity was compromised by infection (a rare occurrence) and the probe was likely broken before explantation. On six further occasions, probes stopped working due to connection errors (typically revealed by a 'shift register' error in SpikeGLX). The recovery rate is therefore 86% when including all connection errors. However, as this type of error is also observed with acute probe use, and there was no observable damage to the chronic probes, these failures may reflect long-term wear rather than any issue with the implant. Consistent with this, probes that failed with this error had typically been used for several months (*Supplementary file 1*). Outside of the originating laboratory (UCL), 95% of probes (19/20) were recovered without any broken shanks (90% when including all connection errors) demonstrating the ease with which new users adopt this design.

## Stability

We tested the stability of the Apollo implant with Neuropixels 1.0, 2.0α, and 2.0 probes (*Figure 3*). We implanted 48 mice using 4-shank Neuropixels 2.0α implants (20 mice with a single-probe implant and 13 mice with a dual implant), single-probe Neuropixels 2.0 implants (7 mice), and Neuropixels 1.0 implants (7 mice with a single-probe implant and one mouse with a dual), as well as 3 rats with a single Neuropixels 1.0 implant (*Supplementary file 1*). In many cases, the same implants were reused (up to six times) and remained fully functional across different animals (*Supplementary file 1*). Recordings were performed over a period of days to months. The probes were typically inserted 5–6 mm inside the brain, traversing multiple brain regions (*Figure 3A, B*). Because only 384 of the 5120 channels, termed a 'bank' of channels, can be recorded simultaneously on each 4-shank 2.0 probe, multiple recording sessions were often used to cover all recording sites located in the brain. Compared with acute recordings, this strategy dramatically reduces experimental setup time and complexity, and is especially beneficial for whole brain approaches. The raw signal quality did not seem to change across

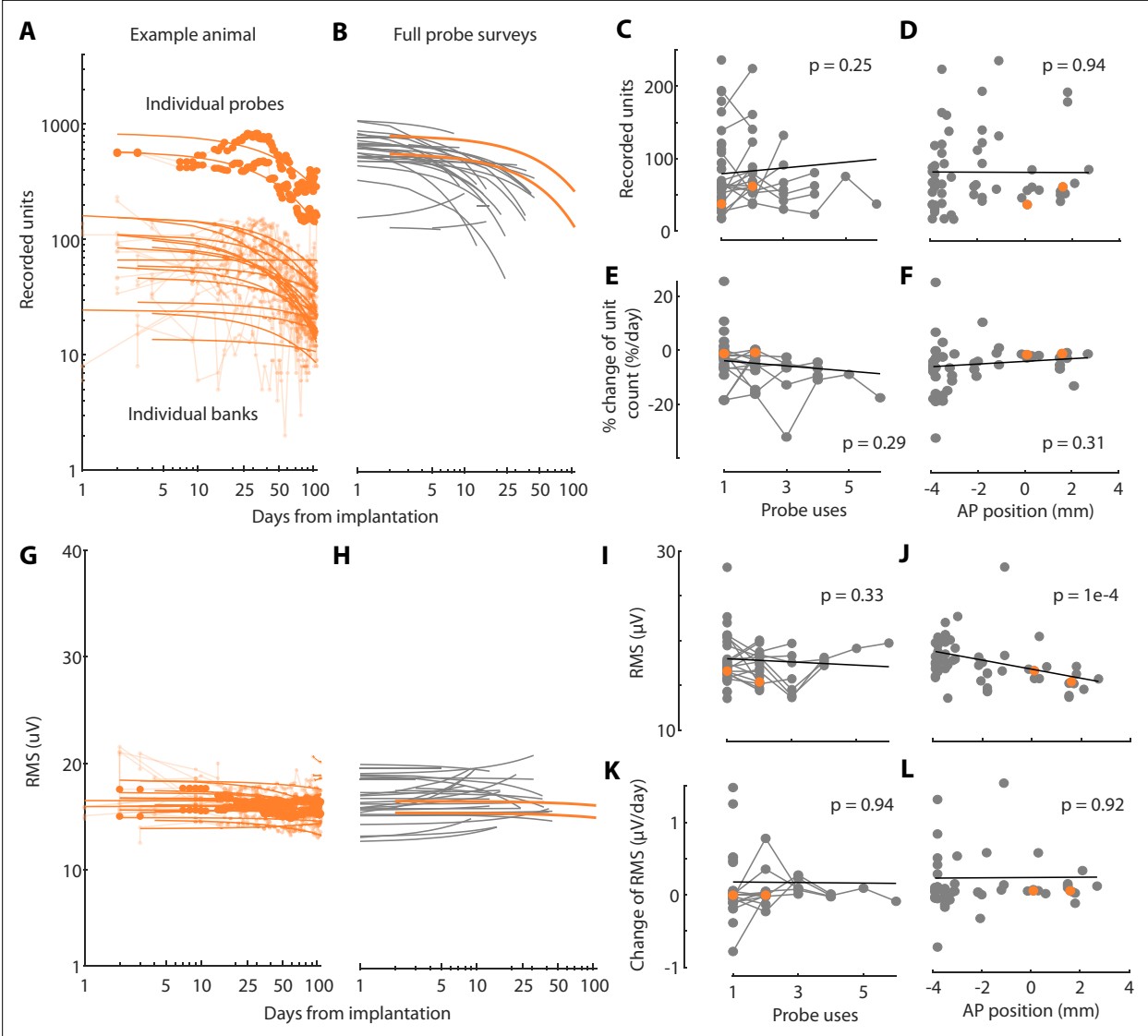

**Figure 4.** Recording stability and implant reuse. (**A**) Total number of recorded units across days for individual channel banks (thin lines), and across each probe (thick lines), for the same implantation as in *Figure 3*. Lines: logarithmic fits. (**B**) Logarithmic fits across all implantations where a full survey of the probe was regularly performed (orange, implantation from *Figure 3*). Full probe surveys were performed only in the primary lab (head-fixed conditions). (**C**) Unit count versus number of implantations. Connected dots represent single probes, reused up to six times. No criteria were used to select probes for reuse, and this decision was based solely on probe availability and experimental need. Slopes were quantified on individual banks and averaged for each probe before applying a linear mixed-effects model (thick line). (**D**) Unit count versus antero-posterior position of the insertion, relative to bregma. (**E, F**) Same as (C, D) but for the slope of the unit count decay. (**G–L**) Same as (A–F) but for the root-mean-square (RMS) value of the raw signal. For **C–F** and **I–L**, all mice are used and shown (head-fixed and freely moving conditions). Rats were excluded because their insertion coordinates cannot be matched with the mice, but their individual results are shown in *Figure 4—figure supplement 1*. All p-values shown come from a linear mixed-effects model.

The online version of this article includes the following figure supplement(s) for figure 4:

**Figure supplement 1.** The amplitude of the recorded neurons is stable across days and probe reuses.

**Figure supplement 2.** Stability of unit count across days.

**Figure supplement 3.** Recording stability on the mouse where the craniotomy was covered with silicon.

**Figure supplement 4.** The amplitude of the recorded neurons is stable across days and probe reuses.

days (*Figure 3C, D*), allowing us to identify single spikes reliably for months. The spiking patterns on each probe were similar across days (*Figure 3E, F*), suggesting that the same populations of neurons were being tracked.

The number of recorded neurons was reasonably stable across weeks (*Figure 4*, *Figure 4—figure supplements 1–3*). For each session, we quantified the number of well-isolated single units for each individual channel bank (*Figure 4A*). Units were selected based on stringent criteria including amplitude, percentage of missing spikes, and refractory period violations (*Fabre et al., 2023*; *van Beest et al., 2024*). The number of single units for each probe is the sum of units across all banks within the brain (*Figure 4A*). Unit numbers could remain stable for more than 50 days, and we observed comparable stability in most of mice (*Figure 4B*). As previously described (*Luo et al., 2020*), we often observed an initial fast decrease in the number of units, but this was not systematic. Indeed, in some animals, unit number increased slowly across days until reaching a peak. The mean decrease in unit count per day was 3% (median 2%), within the range previously observed for chronic Neuropixels implants (*Steinmetz et al., 2021*). Although implants with more rapid unit loss were not suited for long-term recordings, others remained stable for months. Across all banks, the average number of recorded neurons on each bank was 85 ± 6 during the first 10 days (*n* = 59 probes, mean ± SEM), 65 ± 7 during days 10–50 (*n* = 50 probes), 54 ± 15 during days 50–100 (*n* = 6 probes), and 44 ± 25 beyond (*n* = 2 probes) (*Figure 4—figure supplement 2*). The initial number of units did not depend on the number of times the probe was reimplanted (p > 0.25, linear mixed-effects model, *Figure 4C*) or the insertion coordinates of the probe (p > 0.94, linear mixed-effects model, *Figure 4D*). The rate of unit loss was also independent of these two variables (p > 0.29 and p > 0.31 for probe reuse and AP position, linear mixed-effects model, *Figure 4E, F*). However, implant quality was more variable in posterior brain regions, with instances of rapidly decreasing neuron counts, as previously described (*Luo et al., 2020*). Stability was qualitatively similar across different laboratories (*Figure 4—figure supplement 1*). Surgical optimizations are ongoing, and protecting the craniotomy with silicon may significantly increase recording stability (*Figure 4—figure supplement 3*, *Melin et al., 2024*).

The overall quality of the signal remained high throughout days and probe reuses. We quantified the overall noise present in the recordings by computing the root-mean-square (RMS) value of the raw signal (*Figure 4G–L*). The RMS values were stable across days, across all mice (*Figure 4G, H*). Both the average RMS value and its changes over time were independent of the number of times the probe had been used (*Figure 4I, K*). We observed a significant effect of AP position on the RMS value, but not on its changes over time (*Figure 4J, L*). Similarly, the median unit amplitude was stable and unaffected by probe reuse (*Figure 4—figure supplement 4*).

Individual neurons could be tracked across days and months (*Figure 5*). We used the tracking software UnitMatch to track the same units across days, based on their waveforms (*van Beest et al., 2024*). In a mouse recorded for 100 days, a significant fraction of units could be tracked for months (*Figure 5A, B*). Tracked neurons had stable waveforms over days, as expected from the matching procedure, but also stable inter-spike intervals histograms (ISIHs) (*Figure 5C, D*). These ISIHs were not used to match neurons across days, and their stability therefore strongly suggests the same units were tracked over months. The proportion of units tracked between two recordings decreased as a function of time between the recordings (*Figure 5E*). For two recordings on the same day, 50% of neurons were matched, suggesting an upper limit in neuron tracking with the method we used, likely due to variability in neural activity or conservative choices in software parameters. The proportion of tracked neurons typically decreased to 10% after 32 days, but the rate of decay varied across recordings: with some cases where 20% of neurons were tracked for 64 days, and others dropping to 0% after a week. In all cases, tracked units had consistent ISIHs, as measured by the area under the receiver operating characteristic curve (AUC)—comparing the similarity of ISIHs for tracked versus different units (*Figure 5F*). This indicates the tracking algorithm remains accurate for large intervals between recordings.

## Freely behaving animals

To test whether the Apollo implant could be used in more naturalistic conditions, we recorded from freely behaving mice and rats in various configurations (*Figure 6*). First, to minimize the weight, we recorded from two freely moving mice using either a dual Neuropixels 2.0α implant or a Neuropixels 1.0 implant, with the headstage suspended by its connection cable (*Figure 6A, B*). The mice explored

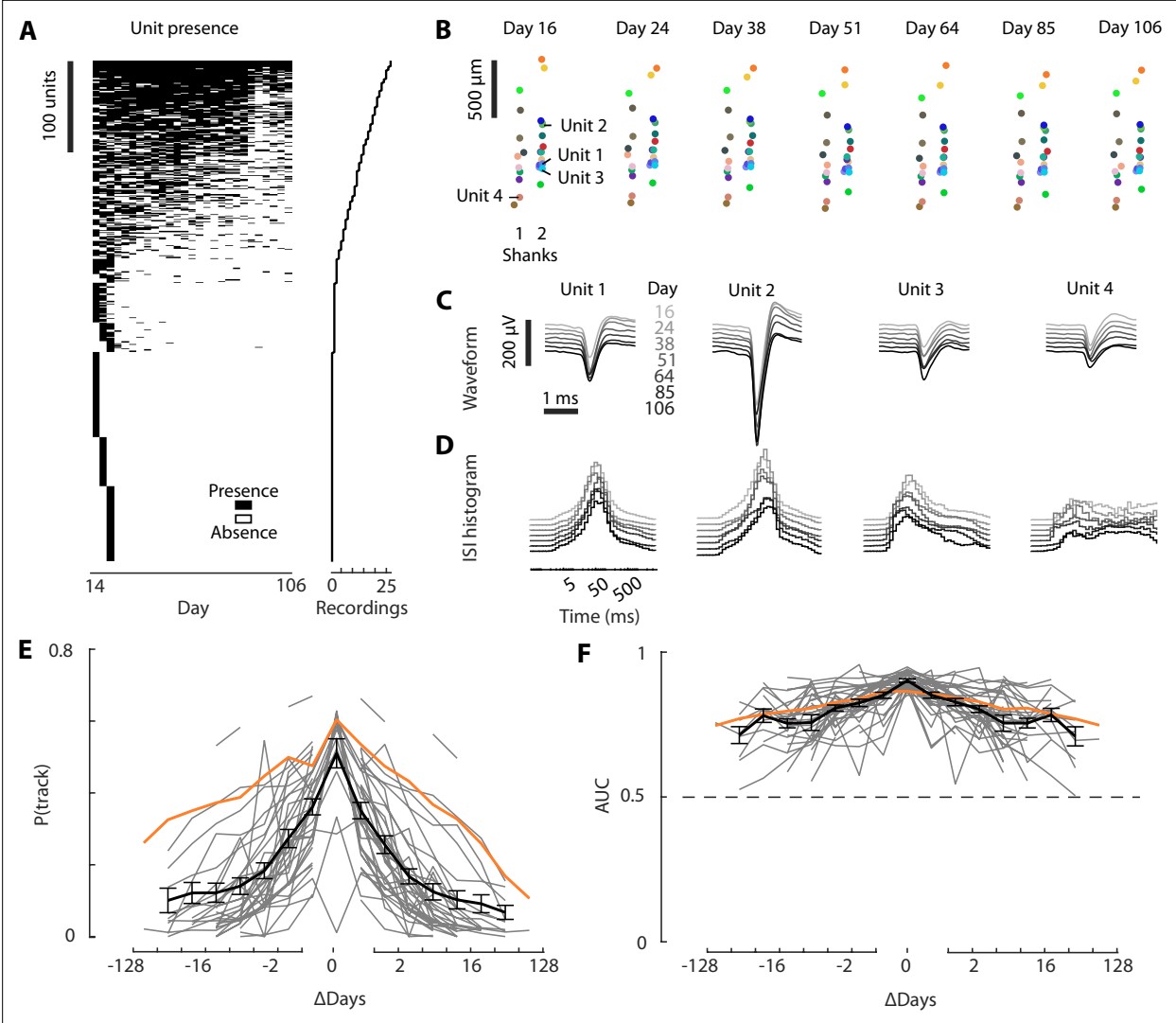

**Figure 5.** Neurons recorded across days and months. (**A**) Unit presence across all recordings (left). Only the units present in at least one of the first 3 days are shown for ease of visualization. Units are ordered by the number of recordings in which their presence was detected (right). (**B**) Spatial layout of the population of neurons tracked across days. In this example, the recording sites were spanning two shanks of a 4-shank 2.0α probe (green rectangle in *Figure 3E*). (**C**) Average waveforms of four example tracked units, computed for 7 days across a 13-week period. (**D**) As in C, but showing the inter-spike interval (ISI) histogram of each unit on each day. (**E**) Probability of tracking a unit as a function of days between recordings, for individual mice (*gray*), including example from A to D (*orange*), or the average across all mice (*black*, mean ± SEM across datasets). (**F**) The average AUC values when comparing the ISI histogram correlations of tracked versus non-tracked neurons. Colors same as (E).

their home cage and exhibited normal behaviors, such as grooming, running, and sleeping, suggesting that the implant did not impair basic movements. The recordings yielded high-quality, well-isolated single units for weeks (*Figure 6C*, *Figure 4—figure supplement 1*). The distributions of the RMS values (*Figure 6D*) and spike amplitudes (*Figure 6E*) were similar to the head-fixed conditions, suggesting an equivalent quality of recording despite differences in conditions, and labs. It can also be more convenient to secure the position of the headstage in each recording, or permanently attach the headstage to the implant. We thus designed a headstage holder, which we tested with Neuropixels 1.0 (*Figure 6F–J*). To further reduce the weight on the mouse, we also designed a 1-probe version of the implant for Neuropixels 2.0, with a minimal headstage holder (*Figure 6K–O*), inserted at an angle (16 or 25 degrees), at the back of the brain. In rats (*Figure 6P–T*), the implant was inserted in the center of a 3D casing, that afforded extra protection, and neural data was recorded wirelessly using SpikeGadgets.

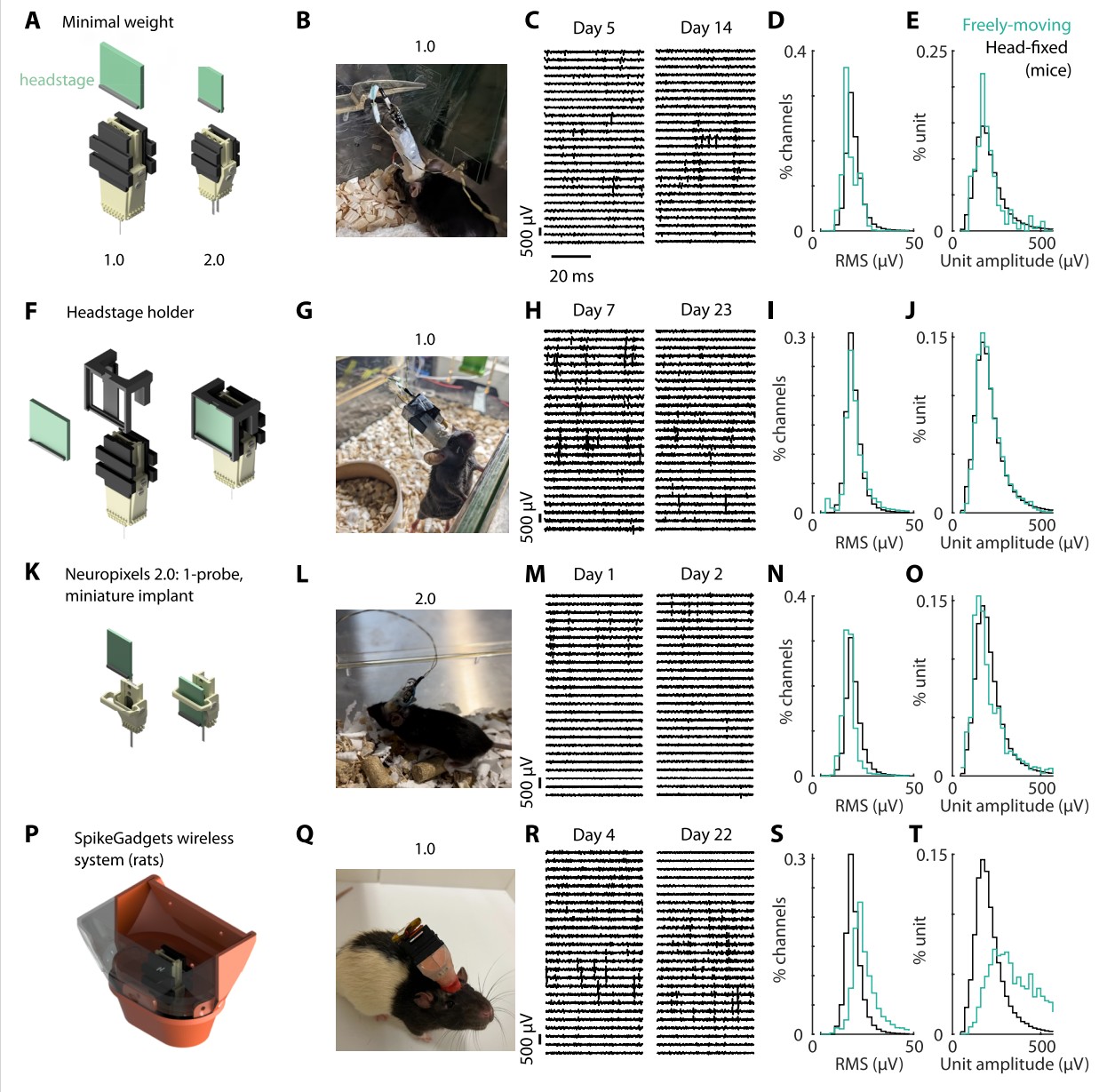

**Figure 6.** Use in freely behaving animals. (**A**) Neuropixels 1.0 and 2.0α were used with freely moving mice. The headstage was suspended by the wire above the implant. (**B**) Animal freely moving with the 1.0 version of the implant, with headstage attached. (**C**) Raw signal (bandpass filtered between 400 Hz and 9 kHz) across multiple channels of increasing depth, on 2 days post-implantation. (**D**) Mean distribution of the root-mean-square (RMS) value across channels, averaged across all recordings in head-fixed mice (black, $n = 2$ mice) and freely moving (cyan, $n = 35$ mice). (**E**) Same as (**D**), but for the distribution of the units' amplitude. (**F**) As in (**A**), but with an additional headstage holder for Neuropixels 1.0 ($n = 4$ mice). (**G–J**) As in (**B–E**) but for recordings with the headstage holder from (D). (**K**) Miniature, 1-probe implant for Neuropixels 2.0, with a headstage holder ($n = 8$ mice with both 2.0 and 2.0α probes). (**L–O**) As in (**B–E**) but using the modified design from (**G**). (**P**) Configuration for rats, with a casing to protect the implant (SpikeGadgets – without the lid, $n = 3$ rats). The final configuration comprises the wireless recording system. (**Q–T**) As in (**B–E**) but with the apparatus from (**P**), recorded in rats. Not that the reference head-fixed data is from mice.

To quantify the effect of implantation on behavior, we compared the performance of mice on a complex behavioral task before and after implantation with a Neuropixels 1.0 probe—the heaviest version of the Apollo Implant (*Table 1*). This implant was modified to allow the headstage to be permanently attached from the first recording session (see Methods). Mice were placed in a large octagonal arena (80 cm diameter). On each trial, mice were required to respond to visual stimuli projected onto the floor of the chamber and perform a nose poke in one of the ports located around the perimeter

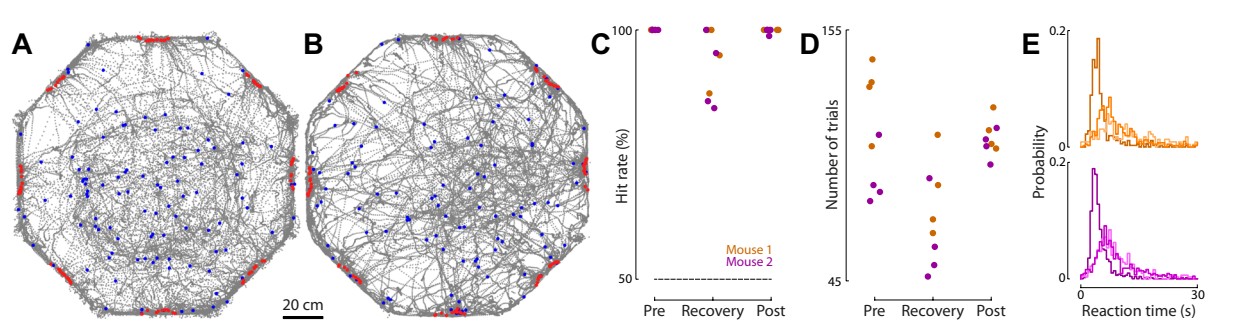

**Figure 7.** Effects of implantation on performance in a freely moving task. (**A**) Trajectories (gray) of single mouse in an example session prior to implantation. Blue and red dots indicate mouse position at the start and end of each trial. (**B**) As in (**A**), but for an example session post-implantation. (**C**) The hit rate (percentage of correctly completed choices) made by two mice (magenta and brown) in the four sessions immediately before implantation (Pre), immediately after implantation (Recovery), and the subsequent four sessions after this recovery period (Post). n = 2 mice, 4 sessions per mouse at each time period. (**D**) As in (**C**), but for the number of trials per session. (**E**) The probability of each reaction time for the same sessions in A and B, separated by mouse.

of the chamber. Thus, during a typical session of 100 trials, mice typically traversed tens of meters (*Figure 7A*). We compared mouse hit rate, trial number, and reaction times in sessions before implantation (when the mouse moved entirely freely) and after implantation (when the mouse was tethered). After implantation, mice continued to perform trials and fully explore the chamber (*Figure 7B*). We observed an initial reduction in hit rate and trial number, and an increase in reaction time immediately after implantation (*Figure 7C–E*). The first two measures recovered within five sessions (*Figure 7C, D*), but reaction times did not recover to pre-implantation levels, indicating that implantation may impact mobility in physically demanding tasks. Although we cannot disambiguate whether these changes are due to tethering, or the implant, it represents the maximal impact of the implantation, particularly as the heaviest Apollo implant was used. Therefore, the consistent hit rate and trial number, in a complex task requiring exploration in a large arena, demonstrates that the implant is well-suited to extended recordings from freely moving mice.

## Discussion

To record large populations of neurons across days and during freely moving behaviors we developed the 'Apollo implant': a chronic implant for Neuropixels 1.0 and 2.0 probes. This solution is easily implanted and recovered, inexpensive, lightweight, flexible, and stable. We successfully tested the implant across multiple labs, setups (head-fixed or freely moving), and species (mice and rats), recording neural populations across weeks and even months.

The design of the Apollo implant builds upon past advances in chronic devices for Neuropixels probes (*Juavinett et al., 2019*; *Luo et al., 2020*; *Steinmetz et al., 2021*; *van Daal et al., 2021*; *Vöröslakos et al., 2021*) to improve on several aspects: weight, price, flexibility, and ease of use. The implant is optimized for animals that cannot carry heavy loads, like mice and especially female and water-controlled mice, which have lower body weight. Because the headstage is not permanently fixed to the implant, the animal carries less weight outside of recordings, and a single headstage can be used with multiple animals in sequence. However, the flexible design allows for the headstage to be permanently attached to an implant, which increases experimental ease at the expense of some additional implant weight. The implant is strong enough to be carried by rats with the addition of a protective 3D outer casing (see Methods), but its use in stronger animals, like ferrets or primates, remains untested. For applications requiring even lighter implants, such as birds, printing materials can be selected to further reduce weight. The lightweight design enables animals to perform complex and demanding freely moving tasks, but also allows experimenters to implant female and water-restricted mice while respecting animal welfare weight limitations.

The Apollo implant is more flexible than previously published solutions. A unique aspect of our modular design is that different docking modules can be used when reimplanting the same payload module, which enables a variety of recording configurations (brain regions, animals, and

experimental setups) that would not have been possible with previous designs. The provided CAD files are fully editable and open source, allowing experienced users to modify the parts as needed. For inexpert users, the files are populated with predefined key dimensions that can be easily adjusted to accommodate changes in several features, including inter-probe distance, angle of implantation, and the length of exposed probes. This ensures the implant remains close to the skull for each experiment, minimizing surgical complications, implant weight (less bonding agent is needed), and moment of inertia (height is minimized). Indeed, even with the heavier Neuropixels 1.0 implants, freely moving mice maintained consistent performance on a complex task after implantation.

Although adapting the design to other commercially available silicon probes is beyond the scope of this study, the flexible design paves the way for future adaptations by individual groups. Because of the low component-cost ($3 per docking module), testing custom modifications is also more cost-effective than with previous solutions. This combination of flexibility and affordability is exemplified by the modifications already used across the eight labs providing data for this manuscript.

With the Apollo implant, the number of recorded neurons exhibited good stability across days, regardless of the number of times the probe had been reimplanted. To provide a realistic estimate for the number of high-quality units that could be recorded across days, we used stringent quality metrics based on unit waveform and spiking properties. Predicting the stability of an implantation was difficult and did not seem to correlate strongly with the quality of surgery (e.g. a small bleed during craniotomy, or ease of probe insertion). We observed more variable stability at the back of the brain, especially in superior colliculus, possibly due to the mechanical constraints imposed during head movements.

The Apollo implant allows for the insertion of up to two parallel probes simultaneously. This can be advantageous: it simplifies surgeries by reducing insertion time and allows probes to be placed in close proximity. However, some users may need to insert multiple probes at different angles. In this case, we are aware of two implant solutions in development that could be better suited, although to our knowledge these remain untested outside the authors' own groups and have only been used in mice (*A Aery Jones, 2023*; *Melin et al., 2024*).

We have demonstrated that neurons recorded with the Apollo implant can be effectively tracked across days, consistent with previous characterizations of chronic Neuropixels implants (*Steinmetz et al., 2021*; *van Beest et al., 2024*). van Beest et al. provide further evidence of neurons tracked with the Apollo implant, and a rigorous quantification of the number of neurons that one can expect to track with these methods. We expect the success of these methods to vary across model systems due to differences in waveform properties—for example, we observed qualitatively higher unit amplitudes in rats in this study. The ability to track neurons across these timescales promises to enhance our understanding of cognitive processes that evolve over long timescales, such as learning or aging.

Overall, the Apollo implant fills an important need to facilitate chronic electrophysiology with Neuropixels probes, particularly in small animals. The simplicity and flexibility of its design are exemplified by the eight independent groups that have successfully used the implant and contributed data to this manuscript.

## Methods

Experimental procedures at UCL and University of Edinburgh were conducted according to the UK Animals Scientific Procedures Act (1986), the European Directives 86/609/EEC and 2010/63/EU on the protection of animals used for experimental purposes, and the Animal Welfare and Ethical Review Body (AWERB). Procedures were conducted under personal and project licenses released by the Home Office following appropriate ethics review.

Experimental procedures at UCLA conformed to the guidelines established by the National Institutes of Health and were approved by the Institutional Animal Care and Use Committee of the University of California, Los Angeles David Geffen School of Medicine.

Experimental procedures at Champalimaud were approved and performed in accordance with the Champalimaud Centre for the Unknown Ethics Committee guidelines and by the Portuguese Veterinary General Board (Direção-Geral de Veterinária, approval 0412/2022).

## Implant design and materials

All parts of the implant (except the constructor probes, Thorlabs) were designed using Autodesk Inventor Professional 2023 software, acquired free of charge through the renewable education plan. Parts were 3D-printed by external companies (primarily SGD 3D, https://sgd3d.co.uk/), or at the SWC FabLab. Stereolithography (SLA, using Rigid4000 resin, Formlabs) was typically used for the payload and docking modules, the docking holder, and the constructor head. Selective laser sintering (using Nylon PA12) was typically used for the payload module lids and payload holder. Brass threaded inserts were manually added to the payload module, payload holder, and docking holder using a soldering iron after printing. For parts (e.g. the payload and docking modules) where strength and inflexibility were advantageous, we used Rigid4000 resin, although this material is denser than Nylon PA12. With this combination, the Neuropixels 1.0, 2.0α, and 2.0 implants weigh 1.7, 1.3, and 0.9 g. The weight of the implants can be further reduced to 1.5, 1.1, and 0.8 g if all parts are printed with Nylon PA12. The full-PA12 implants have been successfully used with 1.0 probes, but remains untested with the 2.0 versions. The miniaturized Neuropixels 2.0 implant with headstage holder weighed 0.6 g by itself or 1.1 g with the probe epoxied and ground attached. All probes used, and any resulting issues/break-ages are detailed in *Supplementary file 1*. Damage resulting from historical procedural steps that are no-longer used (e.g. manually separating the shanks of a 4-shank probe with a needle, now achieved by de-ionized water or strong solvent) or carelessness during probe handling outside of mounting, implantation and explantation are not indicated in the table as they are independent of the implant itself.

In addition to the 3D-printed implant, the following materials are required (due to variable supply, up-to-date links are provided in the GitHub repository):

- M1 Brass knurled inserts (eBay, Aliexpress): To be heat-inserted into the Payload.
- M3 Brass knurled inserts (RS): To be heat-inserted into the PayloadHolder and DockingHolder.
- M1 Screws (Accu): To lock modules together (4 per module).
- M3 5 mm screws (Accu): For connecting Payload Module holder to Thor Labs posts (3 total).
- M3 10 mm screws (Accu): For connecting Constructor Head to Thor Labs posts (3 total) and also for securing the Docking Module in its holder (2–4 depending on your preference).
- M3 20 mm screws (Accu): For tightening the payload module holder (one per holder).
- Mini-series optical posts (Thor labs): 3 for the constructor, potentially more to hold the probes, etc.

## Implant assembly, implantation, and explantation protocol

What follows is the protocol used by the originating laboratory with some minor variants. This is the most thoroughly tested and recommended approach. Methods employed by each individual lab are detailed in a later section.

### Payload module assembly—once per Apollo implant

The payload modules were assembled with either one or two Neuropixels probes. First, all parts were assembled by hand without the probes to ensure a good fit first before fixing the probes permanently to the holder.

1. The probes were sharpened individually using a microgrinder (Narishige EG-45), and an independent holder (*Figure 2—figure supplement 1*). When using only one probe, sharpening can be performed at the end of the assembly, using the payload module holder.
2. The empty payload module was positioned on adhesive putty (Blu Tack) and coated with a thin layer of epoxy (Araldite or Gorilla) or dental cement (Superbond).
3. The probe was affixed and aligned to the payload module, either by eye or using graph paper, before covering the base and electronics with epoxy or dental cement (e.g. Loctite E-60NC HYSOL or Superbond, *Figure 2A*).
4. In the case of a dual implant, the second probe was similarly affixed on the other side of the payload module (*Figure 2B*). The relative position of each probe was adjusted by eye to achieve the required combination of depths.
5. The ground and reference were shorted by connecting them with a silver wire. Occasionally, this wire was then soldered to a socket connector to flexibly connect the ground to a bone screw.

6. For each probe, the flex cable was folded and inserted into the slot on the inside of the lid (*Figure 2B*) before the lid was glued to the payload module with superglue (Loctite) or dental cement (SDI, Wave A1) (*Figure 2A*).
7. Any residual openings were filled in with Kwik-Cast (WPI). When only one probe was used, the back side of the payload was sealed by a lid, with masking tape or a small drop of Kwik-Cast.

## Combining payload and docking modules

For each implantation, a new or previously used payload module was combined with a new docking module. The docking module could be varied between experiments to adjust for variables including insertion depth, angle, or headplate-compatibility.

1. The docking module was positioned in the docking module holder, secured with set screws (M3 10 mm), and slid onto the arms of the constructor.
2. The payload module (with probes) was secured in a payload module holder by tightening the corresponding screw (M3 20 mm) and attached to the end of the constructor with screws (M3 10 mm, *Figure 2C*).
3. The docking module holder was slid along the constructor arms toward the payload module, and the two modules were combined with screws (M1 × 2 mm, *Figure 2D*).
4. Any open gaps were closed with Kwik-Cast (*Figure 2E*).
5. Before most implantations, probes were coated with DiI (Vybrant V22888 or V22885, Thermo Fisher) or DiD (Vybrant V22887, Thermo Fisher) by either manually brushing each probe with a droplet of DiI or dipping them directly into the solution (*Figure 2—figure supplement 1B*).

## Implantation

Craniotomies were performed on the day of the implantation, under isoflurane (1–3% in $O_2$) anesthesia, and after injection of appropriate analgesia and anti-inflammatory drugs (usually Colvasone and Carprofren). Headplate surgery was performed in most cases, either days before or on the same day. The eyes of the animal were protected throughout surgery using eye lubricant.

1. The skull was cleaned, scored (to improve cement adhesion), and leveled. One craniotomy per probe was then performed using a drill or a biopsy punch (*Figure 2F*). Craniotomies were as small as possible (<0.5 mm for single shank, 1 mm of length for 4 shanks), while still allowing for room to adjust probe placement (particularly important with dual, 4-shank probe implants).
2. The exposed brain was covered with Dural-Gel (Cambridge Neurotech) which was allowed to cure for 15–30 min. This step can be performed after probe insertion *if* the brain is suffused with saline throughout insertion.
3. (Optional) A skull screw was inserted into the skull for grounding during recordings.
4. The assembled implant—secured within a payload holder—was positioned using a micromanipulator (Sensapex). After positioning the probe shanks at the surface of the brain (*Figure 2G*), avoiding blood vessels, probes were inserted at slow speed (3–5 μm/s).
5. Once the desired depth was reached (generally when the docking module touched the skull), the implant was sealed using cement (3M RelyX Unicem 2 Automix) (*Figure 2H*). Occasionally, silicone gel or Kwik-Sil was on top of the Dural-Gel to improve stability. Preliminary data from one mouse where this was performed shows excellent stability (*Figure 4—figure supplement 3*).
6. (Optional) The skull screw was connected to the probe's reference/ground wire and secured with a drop of cement. During head-fixed experiments, the animal's headplate can be used for grounding, and combined with internal referencing provided by the Neuropixels, yielded low-noise recordings.
7. Probe function, and in some cases position in the brain, were confirmed by plugging the probes into the acquisition system and visually inspecting the signals.
8. The docking module, skull screw (optional), and skull were covered with Super-Bond polymer, taking care to ensure the payload module (and screws) were not cemented.
9. (Optional) In some cases, a headstage was combined with the implant. In this case, a cap/cover was fitted to the implant to protect and hold the headstage. A connection between the headstage and acquisition hardware was confirmed before removing the animal from anesthesia.

10. (Optional) In rats, a SpikeGadgets targeting cone shielding was assembled around the Apollo implant and affixed to the skull with self-curing orthodontic resin (Ortho-jet, Lang Dental). The probe flex cable was connected to the Spike Gadgets interface board and screws were placed to hold the interface board to the targeting cone assembly, completing the implantation.
11. At the end of the surgery, the animal was given analgesia (Metacam), and allowed to awaken on the heating mat before being placed in a heated recovery box and then returned to its home cage.

## Explantation

Explantations were performed under light isoflurane anesthesia (1–3% in $O_2$).

1. With a micromanipulator, a payload holder was aligned to the payload module, slid into place, and secured with a screw. Ensure the animal's head is well aligned to avoid any friction when retracting the probe.
2. The headstage (if present) was disconnected, and any residual silicon was removed. If a skull screw was used, the attached ground wire was unplugged or cut.
3. Saline was applied to the implant to soften any potential debris or tissue regrowth.
4. The screws between the payload and the docking modules were removed.
5. The payload module was retracted from the docking module using the micromanipulator. In cases where the implant was stiff, a needle or the tip of forceps was used to gently separate the two modules, to hold the docking module in place while the payload was retracted.
6. Once extracted, probes were sometimes contaminated with debris, like Dural-Gel or biological tissue (*Figure 2—figure supplement 1C*). Extensive cleaning with a Tergazyme solution (24 hr) followed by de-ionized water typically cleaned the probes. If this procedure proved insufficient, a 24-hr bath in a stronger detergent (DOWSIL DS-2025 Silicone cleaning solvent) removed the residual tissue. Neither process altered the signal quality.

## Lab-specific methods

## Payload module assembly

### Carandini-Harris laboratory

Number/type of probes: One or two 4-shank Neuropixels 2.0α probes
Shank alignment: By eye
Grounding preparation: Silver wire to short the ground and reference, connected to headplate.
Headstage: Removable

### Churchland laboratory

Number/type of probes: Two Neuropixels 1.0 probes
Sharpening: Yes (Narishige EG-45)
Shank alignment: We checked probe alignment in the holder without any cement or glue. Kapton tape was added to the backside of the probes and then alignment was rechecked to ensure a straight shank trajectory. After verifying proper alignment, the probes were affixed to the payload module, before covering the base and electronics with light curable dental cement (SDI, Wave A1).
Grounding preparation: Silver wire to short the ground and reference, and soldered to a socket connector (Digikey, ED5164-15-ND) to allow for grounding with a bone screw. A bone screw tethered to its own socket connector was also prepared to allow for detachment of the grounding screw from the probes at the conclusion of the experiment.
Headstage: Removable

### Duan laboratory

Number/type of probes: One Neuropixels 1.0 probe
Sharpening: Yes (Narishige EG-45)
Shank alignment: By eye

Grounding preparation: Silver wire to short the ground and reference and soldered to a grounding wire terminating in a gold socket.

Headstage: Permanent, but recoverable. A Neuropixels 1.0 headstage was covered in epoxy resin. After allowing for the epoxy to fully cure, the headstage was attached to the 3D-printed headstage holder using set of screw-holes/threads on the holder.

### Kullman/Lignani laboratories

Number/type of probes: One Neuropixels 1.0 probe
Sharpening: Yes (Narishige EG-45)
Shank alignment: By eye
Grounding preparation: Silver wire to short the ground and reference, and soldered to an additional silver wire, terminated with a male Mill-Max pin and insulated with Plastidip, for connecting to a skull ground screw during recordings.
Headstage: Removable

### Mainen laboratory

Number/type of probes: One Neuropixels 1.0 probe
Sharpening: Yes (Narishige EG-45)
Shank alignment: Using graph paper
Grounding preparation: Silver wire to short the ground and reference.
Headstage: Removable

### Margrie laboratory

Number/type of probes: One 4-shank Neuropixels 2.0α or 2.0 probe
Sharpening: Unsharpened
Shank alignment: By eye
Grounding preparation: Silver wire to short the ground and reference.
Headstage: Removable

### Rochefort laboratory

Number/type of probes: One Neuropixels 1.0 probe
Sharpening: Unsharpened
Shank alignment: By eye
Grounding preparation: Silver wire to short the ground and reference.
Headstage: Removable
Note: To make the design compatible with existing headplates (*Osborne and Dudman, 2014*), the docking module surface was sliced before printing (44° angle).

### Wikenheiser laboratory

Number/type of probes: One Neuropixels 1.0 probe
Sharpening: Unsharpened
Shank alignment: Using graph paper
Grounding preparation: Silver wire to short the ground and reference, and a male gold pin (AM systems catalog #520200) connected to a length of the same silver wire was soldered to a different ground pad on the flex cable. A stainless-steel ground screw (McMaster-Carr catalog #92470A015) was prepared by wrapping a length of silver wire several times below the screw head, affixing the wire to the screw with solder, and soldering the free end of the wire to a female gold pin compatible with the one affixed to the probe.
Headstage: Removable (SpikeGagdets)

Note: The point at which the probe emerged from the docking module was carefully coated with silicone gel (Dow-Corning 1597418).

## Implantation

### Carandini-Harris laboratory

Animals: Adult male and female mice C57BL\6J

Preparatory surgery timing: Different day than implantation

Preparatory surgery: A brief (approximately 1 hr) initial surgery was performed to implant a titanium headplate (approximately 25 × 3 × 0.5 mm, 0.2 g). In brief, the dorsal surface of the skull was cleared of skin and periosteum. A thin layer of cyanoacrylate (VetBond, World Precision Instruments) was applied to the skull and allowed to dry. Thin layers of UV-curing optical glue (Norland Optical Adhesives #81, Norland Products) were applied and cured until the exposed skull was covered. This glue was completely removed during the implantation surgery. The headplate was attached to the skull over the interparietal bone with Super-Bond polymer (Super-Bond C&B, Sun Medical). After recovery, mice were treated with carprofen for 3 days, then acclimated to handling and head-fixation.

Craniotomy methods (size): Biopsy punch or drill, then covered with Dural-Gel (1–1.5 mm)

Insertion speed (manipulator): 3–5 μm/s (Sensapex)

Implant cementing: Blue-light-curing cement (3M RelyX Unicem 2 Automix), then covered with Super-Bond. Only in GB012, the craniotomy was first covered with Kwik-Sil before being sealed.

Treatment: Isoflurane anesthesia, carprofren, and dexamethasone during implantation, then meloxicam or carprofren for 3 days.

### Churchland laboratory

Mice: Adult male mice C57BL\6J

Preparatory surgery timing: Just before implantation

Preparatory surgery: In brief, the dorsal surface of the skull was cleared of skin and periosteum. A thin layer of cyanoacrylate (VetBond, World Precision Instruments) was applied to the edges of skull and allowed to dry. After ensuring the skull was properly aligned within the stereotax, craniotomy locations were marked by making a small etch in the skull with a dental drill. A titanium headbar was then affixed to the back of the skull with a small amount of glue (Zap-a-gap). The headbar and skull were then covered with Metabond, taking care to avoid covering the marked craniotomy locations. After the Metabond was dry, the craniotomies for the probes and grounding screw were drilled.

Craniotomy methods (size): Drill, then covered with Dural-Gel

Insertion speed (manipulator): 5 μm/s (Neurostar)

Implant cementing: Sealed with Kwik-Sil and UV cement (SDI, Wave A1), then covered with Metabond.

Treatment: Isoflurane anesthesia, meloxicam, and enrofloxacin during implantation, then meloxicam and enrofloxacin for 3 days.

### Duan laboratory

Mice: Adult female mice C57BL\6J

Preparatory surgery timing: Just before implantation

Preparatory surgery: In brief, the skull was exposed, cleaned, and aligned in preparation for the implantation. A small craniotomy was performed at the target site. A small well was made around the craniotomy using blue-light-curing dental cement (3M RelyX Unicem 2 Automix). The well was filled with Dural-Gel to set before the implantation. The skin was glued, and the exposed skull was fully covered using dental cement (Superbond, SUN medical). Another small craniotomy (0.5 mm) was performed at the cerebellum, and previously prepared golden pin terminating with a silver wire was inserted and cemented such that the pin was resting on top of the skull.

Craniotomy methods (size): Drill, then covered with Dural-Gel (1 mm)

Insertion speed (manipulator): For the first 100–200 µm the lowering speed was 10–20 µm/s, then reduced to 3–5 µm/s for a subsequent 4000 µm (S-IVM Mini, Scientifica, mounted on the stereotaxic manipulator arm).

Implant cementing: Blue-light-curing dental cement (3M RelyX Unicem 2 Automix). The grounding wire socket was plugged into the implanted grounding pin and cured with Light-curing cement. The implant was wrapped with surgical tape (3M micropore).

Treatment: Isoflurane anesthesia and meloxicam during implantation, then meloxicam for 3 days.

Headstage attachment: After recovery, animals were briefly anesthetized and the surgical tape from the implant was removed. The assembled headstage-holder was attached to the implant using the top set of screw-holes/threads of the payload module of the assembled implant. Implant was wrapped with surgical tape (3M micropore) leaving the headstage connector exposed. The connector was plugged with a matching plug (A79604-001, Omnetics) between the recordings.

## Kullman/Lignani laboratories

Mice: Adult male mice C57BL\6J

Preparatory surgery timing: Just before implantation

Preparatory surgery: In brief, the dorsal surface of the skull was cleared of skin and periosteum. A 0.7-mm burr hole was made on the left (contralateral) parietal skull plate and a skull screw (with 0.5 mm silver wire, terminating at a female Mill-Max pin) was inserted. A thin layer of cyanoacrylate glue (MedBond, Animus Surgical, UK) was applied to the surface of the skull at the sutures to adhere the skull bones together and around the skin perimeter to adhere the surrounding skin to the skull.

Craniotomy methods (size): Biopsy punch (Integral) and bur (R&S Diamond Bur Taper Conical End Super Fine, Z12), then covered with Dural-Gel (1.5 mm)

Insertion speed (manipulator): 3–5 µm/s (Sensapex)

Implant cementing: Blue-light-curing cement (3M RelyX Unicem 2 Automix), then covered with Super-Bond. Once the cement had fully dried, the gold contacts of the probe's ZIF connector were protected with a small strip of parafilm, and another strip of parafilm was wrapped at the top of the implant.

Treatment: Isoflurane anesthesia and buprenorphine during implantation, then meloxicam and amoxicillin.

## Mainen laboratory

Animals: Adult male mice C57BL\6J

Preparatory surgery timing: Just before implantation

Preparatory surgery: In brief, the dorsal surface of the skull was cleared of skin and periosteum. Cyanoacrylate (Vetbond, 3M) was applied between the skull and the remaining scalp. After aligning the skull to the stereotaxic frame, craniotomy coordinates were marked using a lab pen (Nalgene). The skull, excluding the locations for craniotomies, was then covered in a thin layer of Superbond C&B (SunMedical).

Craniotomy methods (size): For the probes, biopsy punch, then covered with Dural-Gel. For the ground screw, drill.

Insertion speed (manipulator): 5 µm/s (Sensapex)

Implant cementing: Blue-light-curing cement (3M RelyX Unicem 2 Automix), then covered with Super-Bond.

Treatment: Carprofen during implantation, and if animals showed reduced motility on a daily basis.

## Margrie laboratory

Animals: Adult male mice C57BL\6J

Preparatory surgery timing: Just before implantation

Preparatory surgery: Mice were separated for at least 2 days prior to surgery. The mouse was secured on a stereotaxic frame (Angle Two, Leica Biosystems). After incision, the skull was roughened, and a small craniotomy was made away from the site of implantation and a ground pin inserted. The ground pin, prepared before surgery, consisted of a gold pin soldered to a short piece of silver wire (0.37 mm diameter) that was inserted into the brain. Next a craniotomy was made, a small well was made around the craniotomy using blue-light-curing cement (3M RelyX Unicem 2 Automix). The dura was removed.

Craniotomy methods (size): Drill, kept wet using saline, then covered with Dural-Gel only after probe insertion (1 mm)

Insertion speed (manipulator): 3 µm/s (Luigs and Neumann)

Implant cementing: Blue-light-curing cement (3M RelyX Unicem 2 Automix), then covered with Super-Bond.

Treatment: Isoflurane anesthesia during surgery. Meloxicam given shortly before surgery and 24 hr after surgery.

## Rochefort laboratory

Animals: Adult male and female mice C57BL\6J

Preparatory surgery timing: At least 3 days before implantation

Preparatory surgery: Headplates (3D-printed RIVETS headplate, 0.54 g, VeroClear resin, 3D Bureau) were implanted during an initial surgery. In brief, the dorsal surface of the skull was cleared of skin and periosteum. A thin layer of cyanoacrylate (VetBond, World Precision Instruments) was applied to the skull and allowed to dry. After ensuring the skull was properly aligned within the stereotax, craniotomy sites were marked by tattoo ink using a sterile pipette tip. Edges of the exposed skull and headplate were scored by a scalpel to improve adhesion, then the headplate was attached to the skull using cyanoacrylate (Super Glue Power Gel, Loctite) and dental cement (Paladur, Heraeus Kuzler). The skull was then covered by another layer of cyanoacrylate (Super Glue Liquid Precision, Loctite) and allowed to dry.

Craniotomy methods (size): Cyanoacrylate on the skull was removed by a hand-held dental drill, and a small craniotomy was drilled, then covered with Dural-Gel (0.1 mm)

Insertion speed (manipulator): 3 µm/s (Sensapex)

Implant cementing: Blue-light-curing cement (3M RelyX Unicem 2 Automix) and dental cement (Paladur, Heraeus Kuzler).

Treatment: Isoflurane anesthesia, buprenorphine, carprofen, and dexamethasone during implantation, then buprenorphine at 24 and 48 hr post-surgery.

## Wikenheiser laboratory

Animals: Long-Evans male and female rats

Preparatory surgery timing: Just before implantation

Preparatory surgery: In brief, the skull was exposed, cleared of connective tissue and periosteum, and leveled. A 1.8-mm trephine (Fine Science catalog #18004-18) was used to lightly inscribe the implant coordinates, and burr holes were drilled to accommodate four stainless-steel anchor screws (McMaster-Carr catalog #92470A015) and the grounding screw (placed above the cerebellum). The craniotomy was drilled using the trephine, exposing the dura mater which was removed using a 27-ga needle and fine forceps.

Craniotomy methods (size): Trephine (1.8 mm)

Insertion speed (manipulator): 1–3 µm/s

Implant cementing: The base of the docking module was previously covered with silicon, then Metabond was applied to the docking module to affix it to the skull and the anchor screws, and the ground pins were connected. Next, the Spike Gadgets targeting cone shielding was assembled around the Apollo implant and affixed to the skull with self-curing orthodontic resin

(Ortho-jet, Lang Dental). The probe flex cable was connected to the Spike Gadgets interface board and screws were placed to hold the interface board to the targeting cone assembly, completing the implantation.

Treatment: Carprofen during implantation, then carprofen for 3 days after surgery, and cephalexin for 14 days after surgery.

## Data acquisition

### Carandini-Harris laboratory

Electrophysiology data acquisition: SpikeGLX (*Karsh, 2022*, versions 20190724, 20201012, 20201103, and 2022101).

Experimental context: During the experiments, mice were typically head-fixed and exposed to sensory stimuli (e.g. visual stimuli such as natural images) or engaged in an audiovisual task (*Coen et al., 2023*) or a visual go-no go task.

Grounding: Headplate (interparietal bone). Internal (electrode tip) reference was sometimes used.

### Churchland laboratory

Electrophysiology data acquisition: SpikeGLX (version 20201103).

Experimental context: During the experiments, mice were head-fixed while performing the IBL task (*Aguillon-Rodriguez et al., 2021*).

Grounding: Bone screw.

### Duan laboratory

Electrophysiology data acquisition: SpikeGLX (version 20230411).

Experimental context: Assisted rotary joint (Doric, AHRJ-OE_1x1_24_HDMI+4) was used to remove cable rotation throughout the recording session. During the recordings, water-restricted mice were tethered and placed in a large (80 cm wide) octagonal arena (built in collaboration with NeuroGears Ltd) and were performing a behavioral task. Briefly, the floor of the arena was used as a projection surface to display the stimuli to the mice. At the bottom of each wall the nose pokes were used to detect mouse responses and deliver water reward. At a random time after the trial onset, two stimuli were presented as a spatial cue on the floor, indicating which wall had an 'active' nose poke with available water reward. To cue the relevant wall, each stimulus was in shape of a triangular 'slice' on the arena floor, with the wide base along the width of the wall that had active pokes, narrowing down to a point at the center. The walls with active ports (2/8 walls) were randomized on trial-by-trial basis, and the distances mice had to take to reach the cued walls depended on their position at the trial onset. Mouse reaction times were measured as the elapsed time from the onset of the stimulus until the registration of the poke-response at a cued wall. Mouse performance (hit rate) was calculated as the percentage of trials mice poked to one of the cued walls within 30 s. If the mouse did not poke within 30 s, the trial was aborted and labeled as a miss trial. Sessions typically lasted 25–30 min (85–100 trials), or until mice began to miss responding to the cued walls.

Mouse position in the arena was recorded with a camera (50 Hz, BFS-U3-16S2M-CS, Flir) and detected online by thresholding of mice against the arena floor, based on the region of interest, size, and intensity (*Lopes et al., 2015*). To increase detectability, IR light strips were added around the arena. Artifacts in tracking due to the tether were filtered out by removing frames with displacements larger than 73.35 cm and using a manual threshold based on the distribution of length of major axis of the tracked object.

Grounding: Gold pin (cerebellum).

### Kullman/Lignani laboratories

Electrophysiology data acquisition: SpikeGLX (versions 20190724 and 20230425).

Experimental context: The animal was briefly anesthetized (4% isoflurane) to connect the probe ground to the skull screw pin, the probe headstage at the ZIF connector, and the recording cable to the headstage at the Omnetics connector. In the first animal, a small loop (5 cm) of recording cable was fastened to the side of the implant using parafilm (*Figure 6A*) to ensure forces exerted on the cable were transmitted to the body of the implant and not the fragile ZIF connector. In the second animal, a custom-made headstage holder (https://github.com/Coen-Lab/chronic-neuropixels/tree/main/XtraModifications/Mouse_FreelyMoving) was 3D-printed to attach to the implant assembly and secured the headstage onto the implant while protecting the ZIF connector. The entirety of the recording cable was attached to a nylon cable using knots of nylon thread spaced at 30-cm intervals along the recording cable. The cable was slack relative to the nylon thread so that the animal could freely explore and rotate without introducing excessive tension and knotting into the cable (*A Aery Jones, 2023*). The animal was allowed to recover and freely navigate its home cage (for implant 1; Techniplast GM500, 39 × 20 cm) or an experimental arena (for implant 2; Techniplast GR900, 40 × 35 cm), located inside a small Faraday container or on the floor of the recording room. During recording sessions, the experimenter rotated the cable to match the animal's rotational movements, to propagate twists along the full length of the cable and reduce localized tangling and cable stress near the implant.

Grounding: Bone screw (contralateral parietal skull plate).

### Mainen laboratory

Electrophysiology data acquisition: SpikeGLX (version 20190413).

Experimental context: The mouse was connected to acquisition hardware and placed in an open field (25 × 33 × 45.5 cm plastic box with a layer of bedding) enclosed in a faraday cage (54 × 41 × 70.5 cm) where it explored freely for 20–40 min. The open field was fitted with an infrared light and video camera (FLIR Chameleon3) to allow monitoring of the animal in the dark and a rotary joint was used to reduce mechanical forces on the recording cable.

Grounding: Bone screw over the cerebellum.

### Margrie laboratory

Electrophysiology data acquisition: SpikeGLX (version 20230815).

Experimental context: We used the same behavioral arena and protocols outlined in *Lenzi et al., 2022*. Each mouse was transferred in their home cage to the experimental room and given at least 5 min to acclimatize to the room under low light conditions. The mouse was then briefly anesthetized and the probe cable was then connected, via a rotary joint to allow cable derotation during recording (Doric, AHRJ-OE_1x1_PT__FC_24), and mice were allowed to explore their home cage to acclimatize to the attachment and recover from anesthesia. Mice were then transferred to the behavioral arena (50 × 20 × 28 cm Perspex box) and the stimulus-presentation monitor (Dell E2210F Black (WSXGA+) 22″ TN) was moved into place.

Grounding: Gold pin near the site of implantation, with silver wire penetrating the brain.

### Rochefort laboratory

Electrophysiology data acquisition: SpikeGLX (version 20230202).

Experimental context: During the experiments, mice were head-fixed and exposed to visual stimuli, that is, drifting gratings and natural movies.

Grounding: Internal (electrode tip) reference was used.

### Wikenheiser laboratory

Electrophysiology data acquisition: Trodes (*Trodes development team, 2023*).

Experimental context: One rat (Wikenheiser001) explored a small square-shaped open-field arena (50 cm/side) and data was acquired using Trodes (Spike Gadgets). Wireless

electrophysiological data was acquired from rats (Wikenheiser002 and Wikenheiser003) as they performed a behavioral task in a circular arena (80 cm diameter) for 40 min.
Grounding: Bone screw (cerebellum).

## Data processing

Sessions were automatically spike-sorted using pyKilosort (**Banga et al., 2022**), python port of Kilosort (**Pachitariu et al., 2016**; version 2.0), and automatically curated using Bombcell (**Fabre et al., 2023**). One mouse (GB012) was spike-sorted using Kilosort 4 (**Pachitariu et al., 2024**).

A variety of parameters were used to select high-quality units, based either on their waveform and their spiking properties.

The template waveform-based criteria were: (1) a maximum of two peaks and one trough, (2) a spatial decay slope below −3 µV·µm⁻¹, defined as the slope of a linear fit (using the MATLAB *polyfit* function) between the maximum absolute amplitude of the peak channel and nearest five channels along the length of the probe (i.e. 75 µm away for Neuropixels 2.0), (3) a duration between 0.1 and 0.8 ms (**Deligkaris et al., 2016**), and (4) fluctuations during baseline not exceeding 30% of the peak amplitude. The raw waveform-based criteria, computed using at least 1000 randomly sampled spikes, were: (1) a minimum mean amplitude of 20 µV (only 1% of units had amplitudes below 30 µV, and increasing this threshold to 50 µV did not affect the results), and (2) a minimum mean signal-to-noise ratio of 0.1, defined as the absolute maximum value divided by the baseline variance. Both somatic and non-somatic spikes (**Deligkaris et al., 2016**) were kept.

The spiking properties-based criteria were: (1) a minimum of 300 spikes, (2) less than 20% of spikes missing, estimated by fitting a Gaussian to the spike amplitude distribution with an additional cut-off parameter below which no spikes are present, (3) a maximum of 10% refractory period violations, using a previously published approach (**Hill et al., 2011**), defining the censored period as 0.1 ms and estimating the refractory period using a window between 0.5 and 10 ms, and (4) a minimum presence ratio of 0.2, defined as the fraction of 1 min bins with at least one spike.

## Data analysis

Raw traces (**Figures 3C, D and 6C**) were obtained by bandpass filtering each channel from spikeGLX between 400 and 9000 Hz (using the MATLAB *bandpass* function) and subtracting the median across channels. The RMS value was computed on the processed signal, and the median across all channels was used to summarize each recording.

To obtain the total number of spikes per second at each depth along the probe (**Figure 3E, F**), we summed spikes across all units present within each 20 µm depth bin.

In the case of 4-shank probes, we estimated the number of recorded units for a probe on a given day (**Figure 4A, B**) by summing units from a complete set of independently recorded banks—a set of channel banks that tiled the entirety of the implanted probe—when available. Because each such set was recorded across at least 2 days, we used the closest recordings within a window of 5 days, centered on the day of interest. Days were excluded if it was not possible to form a complete set of banks within a 5-day window.

To obtain $P$ the percentage change in unit count $N_d$ across days $d$ (4C–F), we fit an exponential decay function to the number of units detected on each bank across days and extracted the decay parameter $\tau$:

$$N_d = N_0 10^{\tau d}$$

$P$ was defined as:

$$P = 100 \times \left( \frac{N_{d+1}}{N_d} - 1 \right) = 100 \times \left( 10^\tau - 1 \right)$$

$P$ was averaged across all longitudinally recorded banks from each implantation to obtain a single value. Only banks with at least three recordings were included.

The median of the units' amplitude and the median RMS values were fitted using a linear fit. Similarly, a single value for each implantation was obtained by taking the variable's mean value across all banks.

To estimate the effect of repeated probe use, $U$, and antero-posterior and medio-lateral insertion coordinates ($Y$ and $X$) on a variable of interest (e.g. the percentage change in unit count, or the RMS values) across days, we used a linear mixed-effects model (using the MATLAB *fitlme* function):

$$P \sim 1 + Y + X + U + (1|\text{probeID}) + (U - 1|\text{probeID})$$

Here, $P$ is the response variable, $Y$, $X$, and $U$ are fixed effect terms, and probeID (the probe identity) is the single grouping variable. We then assessed whether $Y$, $X$, or $U$ had a significant effect on the response variable.

To track neurons across days (*Figure 5*), we used UnitMatch, which uses only the neurons' waveform (*van Beest et al., 2024*). We identified cell presence across days using the intermediate algorithms, which have been shown to maximize the probability of tracking neurons while preserving a low false-positive rate. We then identified the neurons that were tracked across 6 arbitrary days spanning the whole recording period. The inter-spike intervals were computed as the distribution of the times between consecutive spikes, binned on a logarithmic scale from 0 to 5 s. As in *van Beest et al., 2024*, to compute the probability of a unit being tracked, we looked at each unit across all recordings and computed the probability of this unit being tracked in previous or subsequent recordings. These probabilities were then averaged across all the units from each animal, and averaged across animals.

To quantify the amount of information present in the distributions of the correlations of the fingerprints, we computed the ROC curve for different populations of pairs (from putative matched units or nonmatched units) across days. We then computed the area under the ROC curve (AUC) to quantify this difference between distributions. Only sessions with at least 20 matched units were considered. Units that had a match within recordings were excluded from this analysis. For each mouse, the AUCs were then averaged across recording locations.

To compute the distributions of RMS values and unit amplitudes (*Figure 6*), we computed these distributions first for single sessions (across channels and units, respectively), and then computed the average distribution across sessions and animals.

## Acknowledgements

This work was supported by the Biological Sciences Research Council (BB/T016639/1 to MC and PC, BB/T007907/1 to NLR), Wellcome Trust (110120/225992 to PC, 205093 to MC and KDH, 204915 to KDH and MC, 219627 to TWM and CAD, 212285 to DMK, and 227065 to CB), the Simons Initiative for the Developing Brain (NLR), The Simons Collaboration on the Global Brain (AKC), the European Research Council under the European Union's Horizon 2020 research and innovation program (694401 to KDH, 866386 to NLR), the European Molecular Biology Organization (ALTF 740-2019 to CB), the National Institutes of Health (U19NS123716, AKC and MM), the Medical Research Council (MR/V034758/1 to DMK and GL and MR/W006804/1 to AMZ), the GOSH/Spark Research Grant (V4422 to GL), the Gatsby Charitable Foundation (GAT3755 to TWM & CAD), the UCLA Chancellor's Animal Research Committee 3R's grant (AW), Whitehall foundation research grant (2021-12-045 to AW), and UK Research and Innovation (EP/Y008804/1 to CAD). FT is supported by the Sainsbury Wellcome Centre PhD program, JMJF is supported by a Wellcome Trust PhD Studentship and AMZ is supported by the Precision Medicine PhD Program. MC holds the GlaxoSmithKline/Fight for Sight Chair in Visual Neuroscience.

## Additional information

### Funding

| Funder | Grant reference number | Author |
| --- | --- | --- |
| Biotechnology and Biological Sciences Research Council | BB/T016639/1 | Matteo Carandini Philip Coen |

| Funder | Grant reference number | Author |
|---|---|---|
| Biotechnology and Biological Sciences Research Council | BB/T007907/1 | Nathalie L Rochefort |
| Wellcome Trust | 10.35802/110120 | Philip Coen |
| Wellcome Trust | 225992 | Philip Coen |
| Wellcome Trust | 10.35802/205093 | Matteo Carandini Kenneth D Harris |
| Wellcome Trust | 10.35802/204915 | Kenneth D Harris Matteo Carandini |
| Wellcome Trust | 10.35802/219627 | Chunyu A Duan Troy W Margrie |
| Wellcome Trust | 10.35802/212285 | Dimitri Michael Kullmann |
| Wellcome Trust | 227065 | Célian Bimbard |
| Simons Initiative for the Developing Brain | | Nathalie L Rochefort |
| Simons Foundation | | Anne K Churchland |
| European Research Council | 694401 | Kenneth D Harris |
| European Research Council | 866386 | Nathalie L Rochefort |
| European Molecular Biology Organization | ALTF 740-2019 | Célian Bimbard |
| National Institute of Health Sciences | U19NS123716 | Anne K Churchland Maxwell D Melin |
| Medical Research Council | MR/V034758/1 | Dimitri Michael Kullmann Gabriele Lignani |
| Medical Research Council | R/W006804/1 | Arthur M Zhang |
| Great Ormond Street Hospital Charity | V4422 | Gabriele Lignani |
| Gatsby Charitable Foundation | GAT3755 | Chunyu A Duan Troy W Margrie |
| UCLA Chancellor's Animal Research Committee 3R's Grant | | Andrew Wikenheiser |
| Whitehall Foundation | 2021-12-045 | Andrew Wikenheiser |
| UK Research and Innovation | EP/Y008804/1 | Chunyu A Duan |

The funders had no role in study design, data collection and interpretation, or the decision to submit the work for publication. For the purpose of Open Access, the authors have applied a CC BY public copyright license to any Author Accepted Manuscript version arising from this submission.

## Author contributions

Célian Bimbard, Conceptualization, Data curation, Software, Formal analysis, Supervision, Funding acquisition, Investigation, Visualization, Methodology, Writing – original draft, Project administration, Writing – review and editing; Flóra Takács, Data curation, Software, Investigation, Methodology, Writing – review and editing; Joana A Catarino, Julie MJ Fabre, Sukriti Gupta, Stephen C Lenzi, Maxwell D Melin, Nathanael O'Neill, James S Street, José M Gomes Teixeira, Enny H van Beest, Arthur M Zhang, Investigation, Writing – review and editing; Ivana Orsolic, Investigation, Visualization, Writing – review and editing; Magdalena Robacha, Simon Townsend, Investigation; Anne K Churchland, Kenneth D Harris, Dimitri Michael Kullmann, Gabriele Lignani, Zachary

F Mainen, Troy W Margrie, Supervision, Funding acquisition; Chunyu A Duan, Nathalie L Rochefort, Andrew Wikenheiser, Supervision, Funding acquisition, Writing – review and editing; Matteo Carandini, Conceptualization, Supervision, Funding acquisition, Writing – review and editing; Philip Coen, Conceptualization, Data curation, Software, Supervision, Funding acquisition, Investigation, Visualization, Methodology, Writing – original draft, Project administration, Writing – review and editing

**Author ORCIDs**
Célian Bimbard ⓘ https://orcid.org/0000-0002-6380-5856
Sukriti Gupta ⓘ https://orcid.org/0009-0004-5222-4775
José M Gomes Teixeira ⓘ https://orcid.org/0000-0003-1787-1809
Enny H van Beest ⓘ https://orcid.org/0000-0002-2454-0445
Anne K Churchland ⓘ https://orcid.org/0000-0002-3205-3794
Kenneth D Harris ⓘ https://orcid.org/0000-0002-5930-6456
Dimitri Michael Kullmann ⓘ https://orcid.org/0000-0001-6696-3545
Gabriele Lignani ⓘ https://orcid.org/0000-0002-3963-9296
Zachary F Mainen ⓘ https://orcid.org/0000-0001-7913-9109
Troy W Margrie ⓘ https://orcid.org/0000-0002-5526-4578
Nathalie L Rochefort ⓘ https://orcid.org/0000-0002-3498-6221
Matteo Carandini ⓘ https://orcid.org/0000-0003-4880-7682
Philip Coen ⓘ https://orcid.org/0000-0003-1495-1061

### Ethics

Experimental procedures at UCL and University of Edinburgh were conducted according to the UK Animals Scientific Procedures Act (1986), the European Directives 86/609/EEC and 2010/63/EU on the protection of animals used for experimental purposes, and the Animal Welfare and Ethical Review Body (AWERB). Procedures were conducted under personal and project licenses released by the Home Office following appropriate ethics review. Experimental procedures at UCLA conformed to the guidelines established by the National Institutes of Health and were approved by the Institutional Animal Care and Use Committee of the University of California, Los Angeles David Geffen School of Medicine. Experimental procedures at Champalimaud were approved and performed in accordance with the Champalimaud Centre for the Unknown Ethics Committee guidelines and by the Portuguese Veterinary General Board (Direção-Geral de Veterinária, approval 0412/2022).

Reviewer #1 (Public review): https://doi.org/10.7554/eLife.98522.3.sa1
Reviewer #2 (Public review): https://doi.org/10.7554/eLife.98522.3.sa2
Reviewer #3 (Public review): https://doi.org/10.7554/eLife.98522.3.sa3
Author response https://doi.org/10.7554/eLife.98522.3.sa4

---

## Additional files

### Supplementary files
MDAR checklist

Supplementary file 1. Table of all animals used in the study. This table provides extensive information on the animals used in the study, such as the implant date, probe number and type, number of recordings, or whether an issue was encountered during the implantation period.

### Data availability
Data is available at https://doi.org/10.6084/m9.figshare.27652932. Note that due to space limits, raw electrophysiological data, along with the spikes data (including what is needed for tracking in *Figure 5*), is only available from the authors upon reasonable requests. The available data contains information about the clusters after spikesorting, including their quality metrics, the RMS of each channel in each recording, and the recordings' metadata. Code to replicate the figures is available at https://github.com/Coen-Lab/chronic-neuropixels/tree/main/Paper-figures (copy archived at *Coen et al., 2024*).

The following dataset was generated:

| Author(s) | Year | Dataset title | Dataset URL | Database and Identifier |
|---|---|---|---|---|
| Ann Duan C, Mainen ZF, Margrie T, Rochefort N, Wikenheiser A, Carandini M, Coen P, Bimbard C, Takacs F, Catarino J, Fabre J, Gupta S, Lenzi S, Melin M, O'Neill N, Orsolic I, Robacha M, Street J, Texeira J, Townsend S, Beest E, Zhang A, Churchland A | 2024 | Dataset for Bimbard et al., 2024 | https://doi.org/10.6084/m9.figshare.27652932 | figshare, 10.6084/m9.figshare.27652932.v1 |

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
