## [Editor Report · eLife Assessment]

This **valuable** study presents the design of a new device for using high-density electrophysiological probes ('Neuropixels') chronically and in freely moving rodents. The evidence demonstrating the system's versatility and ability to record high-quality extracellular data in both mice and rats is **compelling**. This study will be of significant interest to neuroscientists performing chronic electrophysiological recordings.

---

## [Referee Report · Reviewer #1 (Public review)]

Summary:

In this manuscript by Bimbard et al., a new method to perform stable recordings over long periods of time with neuropixels as well as the technical details on how the electrodes can be explanted for a follow up reuse is provided. I think the description of all parts of the method are very clear, and the validation analyses (n of units per day over time, RMS over recording days...) are very convincing. I however missed a stronger emphasis on why this could provide a big impact on the ephys community, by enabling new analyses, new behavior correlation studies or neurophysiological mechanisms across temporal scales that were previously inaccessible with high temporal resolution (i.e. not with imaging).

Strengths:

Open source method. Validation across laboratories. Across species (mice and rats) demonstration of its use and in different behavioral conditions (head-fixed and freely moving). The implant offers a major advance compared to previous methods and that will help the community generate richer datasets.

Weaknesses:

None noted.

---

## [Referee Report · Reviewer #2 (Public review)]

Summary:

This work by Bimbard et al., introduces a new implant for Neuropixels probes. While Neuropixels probes have critically improved and extended our ability to record the activity of a large number of neurons with high temporal resolution, the use of these expensive devices in chronic experiments has so far been hampered by the difficulty of safely implanting them and, importantly, to explant and reuse them after conclusion of the experiment. The authors present a newly designed two-part implant, consisting of a docking and a payload module, that allows for secure implantation and straightforward recovery of the probes. The implant is lightweight, making it amenable for use in mice and rats, and customizable. The authors provide schematics and files for printing of the implants, which can be easily modified and adapted to custom experiments by researchers with little to no design experience. Importantly, the authors demonstrate the successful use of this implant across multiple use cases, in head-fixed and freely moving experiments, in mice and rats, with different versions of Neuropixels probes and across 8 different labs. Taken together, the presented implants promise to make chronic Neuropixels recordings and long-term studies of neuronal activity significantly easier and attainable for both current and future Neuropixels users.

Strengths:

- The implants have been successfully tested across 8 different laboratories, in mice and rats, in head-fixed and freely moving conditions and have been adapted in multiple ways for a number of distinct experiments.

- Implants are easily customizable and authors provide a straightforward approach for customization across multiple design dimensions even for researchers not experienced in design.

- The authors provide clear and straightforward descriptions of the construction, implantation and explant of the described implants.

- The split of the implant into a docking and payload module makes reuse even in different experiments (using different docking modules) easy.

- The authors demonstrate that implants can be re-used multiple times and still allow for high-quality recordings.

- The authors show that the chronic implantations allow for the tracking of individual neurons across days and weeks (using additional software tracking solutions), which is critical for a large number of experiments requiring the description of neuronal activity, e.g. throughout learning processes.

- The authors show that implanted animals can even perform complex behavioral tasks, with no apparent reduction in their performance.

---

## [Referee Report · Reviewer #3 (Public review)]

Summary:

In this manuscript, Bimbard and colleagues describe a new implant apparatus called "Apollo Implant", which should facilitate recording in freely moving rodents (both mice and rats) using Neuropixels probes. The authors collected data from both mice and rats, they used 3 different versions of Neuropixels, multiple labs have already adopted this method, which is impressive. They openly share their CAD designs and surgery protocol to further facilitate the adaptation of their method.

Strengths:

Overall, the "Apollo Implant" is easy to use and adapt, as it has been used in other laboratories successfully and custom modifications are already available. The device is reproducible using common 3D printing services and can be easily modified thanks to its CAD design (the video explaining this is extremely helpful). The weight and price are amazing compared to other systems for rigid silicon probes allowing a wide range of use of the "Apollo Implant".

Weaknesses:

The "Apollo Implant" can only handle Neuropixels probes. It cannot hold other widely used and commercially available silicon probes. Certain angles and distances may be better served by 2 implants.

---

## [Author Response]

The following is the authors’ response to the original reviews.

**Reviewer 1 (Public Review):**
Summary:In this manuscript by Bimbard et al., a new method to perform stable recordings over long periods of time with neuropixels, as well as the technical details on how the electrodes can be explanted for follow-up reuse, is provided. I think the description of all parts of the method is very clear, and the validation analyses (n of units per day over time, RMS over recording days...) are very convincing. I however missed a stronger emphasis on why this could provide a big impact on the ephys community, by enabling new analyses, new behavior correlation studies, or neurophysiological mechanisms across temporal scales.Strengths:Open source method. Validation across laboratories. Across species (mice and rats) demonstration of its use and in different behavioral conditions (head-fixed and freely moving).Weaknesses:

Weak emphasis on what can be enabled with this new method that didn't exist before.

We thank the reviewer for highlighting the limited discussion around scientific impact. Our implant has several advantages which combine to make it much more accessible than previous solutions. This enables a variety of recording configurations that would not have been possible with previous designs, facilitating recordings from a wider range of brain regions, animals, and experimental setups. In short, there are three key advances which we now emphasise in the manuscript:

Adaptability: The CAD files can be readily adapted to a wide range of configurations (implantation depth, angle, position of headstage, etc.). Labs have already modified the design for their needs, and re-shared with the community (Discussion, Para 5).

Weight: Because of the lightweight design, experimenters can (i) perform complex and demanding freely moving tasks as we exemplify in the manuscript, and (ii) implant female and water restricted mice while respecting animal welfare weight limitations (Flexible design, Para 1).

Cost: At ~$10, our implant is significantly cheaper than published alternatives, which makes it affordable to more labs and means that testing modifications is cost-effective (Discussion, Para 4).

**Reviewer 1 (Recommendations For The Authors):**
- Differences between mice and rats seem very significant. Although this is probably not surprising, I suggest that the authors comment on this to make it clear to anyone trying to use in different species that are not quantified in the main figures.

The reviewer is correct—there are qualitative differences between mice and rats, particularly with respect to the unit median amplitude. We have added a comment in the discussion to highlight these inter-species variations (Discussion, Para 7)

- Another comment that would be useful to have would be how to tackle the problem of tracking the same neuron across days. Even if currently impossible, it could be useful to provide discussion along those lines as to where future improvements (either in hardware or software) can be made.

We thank the reviewer for highlighting this. Figure. 5 does show data from tracking the same neuron across days (and even months). We have modified the language to make this clear.

**Reviewer 2 (Public Review):**
Summary:This work by Bimbard et al., introduces a new implant for Neuropixels probes. While Neuropixels probes have critically improved and extended our ability to record the activity of a large number of neurons with high temporal resolution, the use of these expensive devices in chronic experiments has so far been hampered by the difficulty of safely implanting them and, importantly, to explant and reuse them after conclusion of the experiment. The authors present a newly designed two-part implant, consisting of a docking and a payload module, that allows for secure implantation and straightforward recovery of the probes. The implant is lightweight, making it amenable for use in mice and rats, and customizable. The authors provide schematics and files for printing of the implants, which can be easily modified and adapted to custom experiments by researchers with little to no design experience. Importantly, the authors demonstrate the successful use of this implant across multiple use cases, in head-fixed and freely moving experiments, in mice and rats, with different versions of Neuropixels probes, and across 8 different labs. Taken together, the presented implants promise to make chronic Neuropixel recordings and long-term studies of neuronal activity significantly easier and attainable for both current and future Neuropixels users.Strengths:The implants have been successfully tested across 8 different laboratories, in mice and rats, in headfixed and freely moving conditions, and have been adapted in multiple ways for a number of distinct experiments.Implants are easily customizable and the authors provide a straightforward approach for customization across multiple design dimensions even for researchers not experienced in design.The authors provide clear and straightforward descriptions of the construction, implantation, and explant of the described implants.The split of the implant into a docking and payload module makes reuse even in different experiments (using different docking modules) easy.The authors demonstrate that implants can be re-used multiple times and still allow for high-quality recordings.The authors show that the chronic implantations allow for the tracking of individual neurons across days and weeks (using additional software tracking solutions), which is critical for a large number of experiments requiring the description of neuronal activity, e.g. throughout learning processes.The authors show that implanted animals can even perform complex behavioral tasks, with no apparent reduction in their performance.Weaknesses:While implanted animals can still perform complex behavioral tasks, the authors describe that the implants may reduce the animals' mobility, as measured by prolonged reaction times. However, the presented data does not allow us to judge whether this effect is specifically due to the presented implant or whether any implant or just tethering of the animals per se would have the same effects.

The reviewer is correct: some of the differences in mouse reaction time could be due to the tether rather than the implant. As these experiments were also performed in water-restricted female mice with the heavier Neuropixels 1.0 implant, our data represent the maximal impact of the implant, and we have highlighted this point in the revision (Freely behaving animals, Para 2).

While the authors make certain comparisons to other, previously published approaches for chronic implantation and re-use of Neuropixels probes, it is hard to make conclusive comparisons and judge the advantages of the current implant. For example, while the authors emphasize that the lower weight of their implant allows them to perform recordings in mice (and is surely advantageous), the previously described, heavier implants they mention (Steinmetz et al., 2021; van Daal et al., 2021), have also been used in mice. Whether the weight difference makes a difference in practice therefore remains somewhat unclear.

The reviewer is correct: without a direct comparison, we cannot be certain that our smaller, lighter implant improves behavioural results (although this is supported by the literature, e.g. Newman *et al*, 2023). However, the reduced weight of our implant is critical for several laboratories represented in this manuscript due to animal welfare requirements. Indeed, in van Daal *et al* the authors “recommend a [mouse] weight of >25 g for implanting Neuropixels 1.0 probes.” This limit precludes using (the vast majority of) female mice, or water-restricted animals. Conversely, our implant can be routinely used with lighter, water-restricted male and female mice. We emphasised this point in the revision (Discussion, Para 2).

The non-permanent integration of the headstages into the implant, while allowing for the use of the same headstage for multiple animals in parallel, requires repeated connections and does not provide strong protection for the implant. This may especially be an issue for the use in rats, requiring additional protective components as in the presented rat experiments.

We apologise for not clarifying the various headstage holder options in the manuscript and we have now addressed this in the revision (Freely behaving animals, Para 1&2). Our repository has headstage holder designs (in the XtraModifications/Mouse_FreelyMoving folder). This allows leaving the headstage on the implant, and thus minimize the number of connections (albeit increasing the weight for the mouse). Indeed, mice recorded while performing the task described in our manuscript had the head-stage semi-permanently integrated to the implant, and we now highlight this in the revision (Freely behaving animals, Para 1).

**Reviewer 2 (Recommendations For The Authors):**
The description of the different versions of the head-stage holders should be more clear, listing also advantages/disadvantages of the different solutions. It would be also useful if the authors could comment on the use of these head-stage holders in rats, since they do not seem to offer much protection.

We thank the reviewer for this point, and we have added notes to the manuscript to clarify the various advantages of the different headstage-holders, and that the headstage can be permanently attached to the implant (Freely behaving animals, Para 1&2). This is the primary advantage of these solutions compared with the minimal implant—at the expense of increasing the implant weight.

The reviewer’s concerns regarding the lack of protection for implants in rats is well-placed, and we now emphasise that these experiments benefited from the additional protection of an external 3D casing, which is likely critical for use in larger animals (Freely behaving animals, Para 1).

While re-used probes seem to show similar yields across multiple uses (Figure 4C), it seems as if there is a much higher variability of the yield for probes that are used for the first (maybe also second) time. There are probes with much higher than average yields, but it seems none of the re-used probes show such high yields. Is this a real effect? Is this because the high-yield probes happened to have not been used multiple times? Is there an analysis the authors could provide to reduce the concern that yields may generally be lower for re-used probes/that there are no very high yields for re-used probes?

We understand the reviewer’s concern with respect to Figure 4C, however, the re-use of any given probe was determined only by the experimental needs of the project. It is therefore not possible that there is a relationship between probes selected for re-use and unit-yield. We now specify this in the revised legend of Figure 4C. This variability (and the consistency in yield across uses) likely stems from differences between labs, brain region, and implantation protocol.

The authors claim that a 'large fraction' of units could be tracked for the entire duration of the experiment (Figure 5A,B). They mention in the discussion that quantification can be found in a different manuscript (van Beest et al., 2023), but this should also be quantified here in at least some more detail, also for other animals in addition to the one mouse which was recorded for ~100 days. What fraction can be held for different durations? What is the average holding time, etc.?

We agree with the reviewer, and have now added new panels quantifying the probability and reliability of tracking a neuron across days (Figure 5E-F). We also comment on the change in tracking probability across time, and its variability across recordings (Stability, Para 4).

**Reviewer 3 (Public Reviews):**
Summary:In this manuscript, Bimbard and colleagues describe a new implant apparatus called "Apollo Implant", which should facilitate recording in freely moving rodents (mice and rats) using Neuropixels probes. The authors collected data from both mice and rats, they used 3 different versions of Neuropixels, multiple labs have already adopted this method, which is impressive. They openly share their CAD designs and surgery protocol to further facilitate the adaptation of their method.Strengths:Overall, the "Apollo Implant" is easy to use and adapt, as it has been used in other laboratories successfully and custom modifications are already available. The device is reproducible using common 3D printing services and can be easily modified thanks to its CAD design (the video explaining this is extremely helpful). The weight and price are amazing compared to other systems for rigid silicon probes allowing a wide range of use of the "Apollo Implant".Weaknesses:The "Apollo Implant" can only handle Neuropixels probes. It cannot hold other widely used and commercially available silicon probes. Certain angles and distances are not possible in their current form (distance between probes 1.8 to 4mm, implantation depth 2-6.5 mm, or angle of insertion up to 20 degrees).

As we now discuss in the manuscript (Discussion, Para 4), one implant accommodating the diversity of the existing probes is beyond the scope of this project. However, because the design is adaptable, groups should be able to modify the current version of the implant to adapt to their electrodes’ size and format (and can highlight any issues in the Github “Discussions” area).

With Neuropixels, the current range of depths covers practically all trajectories in the mouse brain. In rats, where deeper penetrations may be useful, the experimenter can attach the probe at a lower point in the payload module to expose more of the shank. We now specify this in the Github repository.

We have now extended the range of inter-probe distances from a maximum of 4 mm to 6.5 mm. Distances beyond this may be better served by 2 implants, and smaller distances could be achieved by attaching two probes on the same side of the docking module. These points are now specified in the revised manuscript (Flexible design, Para 2).

**Reviewer 3 (Recommendations For The Authors):**
I have only a few questions and suggestions:Is it possible to create step-by-step instructions for explantation (similar to Figure-1 with CAD schematics)? You mention that payload holder is attached to a micromanipulator, but it is unclear how this is achieved. How was the payload secured with a screw (which screw)? My understanding is that as you turn the screw in the payload holder, it will grab onto the payload module from both sides, but the screw is not in contact with the payload module, correct? I found the screw type on your GitHub, but it would be great if you could add a bill of materials in a table format, so readers don't have to jump between GitHub and article.

We have now added a bill of materials to the revised manuscript (Implant design and materials, Para 2), although up-to-date links are still provided on the Github repository due to changing availability.

What happens if you do a dual probe implant and cannot avoid blood vessels in one or both of the craniotomies due to the pre-defined geometry? Is this a frequent issue? How can you overcome this during the surgery?

Blood vessels can be difficult to avoid in some cases, but we are typically able to rotate/reposition the probes to solve this issue. In some cases, with 4-shank probes, the blood vessel can be positioned between individual probe shanks. We now detail this in the revised manuscript (Assembly and implantation, Para 3).

I assume if the head is not aligned (line-332) the probe can break during recovery. Have you experienced this during explanation?

As we now specify in the manuscript (Explantation, Para 2), we are careful when explanting the probe to avoid this issue, and due to the flexibility of the shanks, it does not appear to be a major concern.

Why did you remove the UV glue (line 435)? How can you level the skull? I assume you have covered bregma and lambda in the first surgery which can create an uneven surface to measure even after you remove the UV glue.

We thank the reviewer for highlighting this omission from the methods. We now explain (Implantation, Carandini-Harris laboratory) that the UV-glue is completely removed during the second surgery, and the skull is cleaned and scored. This improves the adhesion of the dental cement, and allows for reliable levelling of the skull.

In line 112 you mentioned that the number of recorded neurons was stable; however, you found a 3% mean decrease in unit count per day (line 120). Stability is great until day 10 (in Figure 4A), but it deteriorates quickly after that. I think it would help readers if you could add the mean{plus minus}SEM of recorded units in the text for days 1-10, days 11-50, and days 51-100 (using the data from Figure 4A).

We have now added Supplementary Figure 4 to show unit count across bins of days, and a corresponding comment in the text (Stability, Para 2).

A full survey of the probe (Figure 4B) means that you recorded neuronal activity across 4-5000 channels (depending on how many channels were in the brain). While it is clear that a full probe survey can reduce the number of animals needed for a study, it is also clear in this figure that by day 25 you can record ~300 neurons on 4000 channels. It would be great to discuss this in the discussion and give a balanced view of the long-term stability of these recordings.

Overall, keeping a large number of units for a long time still remains a challenge. Here, we could record on average 85 neurons per bank during the first 10 days, and then only 45 after 50 days. It is important to note that our quantification averages across all banks recorded, including those in a ventricle or partly outside of the brain. Thus, our results represent a lower estimate of the total neurons recorded. Our new Supplementary Figure 4 helps to highlight the diversity of neuron number recorded per animal. Further improvements in surgical techniques and spike sorting will likely improve stability further and we have now added this comment in the manuscript (Stability, Para 2). For example, we observed excellent stability in a mouse where the craniotomy was stabilized with KwikSil (Supplementary Figure 5).

The RMS value was around 20 uV in some of the recordings, and according to Figure 4G it is around 16 uV on average. Is it safe to accept putative single units with 20 uV amplitudes, when the baseline noise level is this close to the spike peak-to-peak amplitude?

On average, less than 1% of the units selected using all the other metrics except the amplitude had an amplitude below 30 µV, and 2.6% below 50 µV. Increasing the threshold to 30 µV, or even 50 µV, did not affect the results. We have now added this comment in the Methods (Data processing, Para 3).

Can you add the waveform and ISIH of the example unit from day 106 to Figure 5?

We have now added 4 units tracked up to day 106 in Figure 5.

Could you move Supplementary Figure 3A to Figure 4? The number of units is more valuable information than the RMS noise level. I understand that you don't have such a nice coverage of all the days as in Figure 3 and 4, but you might be able to group for the first 3 days and the last 3 days (and if data is available, the middle 3 days) as a boxplot. The goal would be for the reader to be able to see whether there is any change in the number of single units over time.

We agree with the reviewer, the number of units is more valuable. We had included this information in Figure 4A-F, but we have made edits to the text to make it clearer that this is what is being shown. The data from Figure 3A is already contained within Figure 4, but in 3A the data is separated by individual labs.

Product numbers are missing in multiple places: line-285 (screw), line-288 (screw), line-290 (screw), line-309 (manipulator), line-374 (gold pin and silver wire), line-384 (Mill-Max), line-394 (silver wire), and many more. It would be great if you could add all these details, so people can replicate your protocol.

We thank the reviewer for highlighting this, and we have added details of screw thread-size and length to relevant parts of the manuscript, although any type of screw can be used. Similarly, other components are non-specific (e.g. multiple silver-wire diameters were used across labs), so we have not included specific product numbers for general consumer items (like screws and silver wires) to avoid indicating that a specific part must be purchased.

While it is great to see lab-specific methods, I am not sure in their current form it helps to understand the protocol better. The information is conveyed in different ways (I assume these were written by different people), in different orders, and in different depths (some mention probe implant location relative to bregma and midline, some don't). There are many different glues, epoxies, cement, wires, and pins. I would recommend rewriting these methods sections under a unified template, so it is easier to follow.

We thank the reviewer for this suggestion and we have rewritten this section of the methods accordingly. We now use a template structure to simplify the comparisons between labs: the same template is used for each lab in each section (payload module assembly, implantation, and data acquisition).

Line-307: why is a skull screw optional for grounding? What did you use for ground and reference if not a ground screw?

We now specify in the manuscript that during head-fixed experiments, the animal’s headplate can be used for grounding, and combined with internal referencing provided by the Neuropixels, yielded lownoise recordings (Implantation protocol, Methods).